

# The Arctic Traits Database – A repository of arctic benthic invertebrate traits

Renate Degen[1] & Sarah Faulwetter[2, 3]

[1] Department of Limnology and Bio-Oceanography, University of Vienna, 1090 Vienna, Austria

[2] Department of Zoology, University of Patras, 26504 Rio, Greece

[3] Institute of Oceanography, Hellenic Centre for Marine Research, 19013 Anavyssos, Greece

*Correspondence to:* Renate Degen (renate.degen@hotmail.com)

**Abstract.** The recently increased interest in marine trait-based studies highlights one general demand – the access
to standardized, reference-based trait information. This demand holds especially true for polar regions, where the
gathering of ecological information is still challenging. The Arctic Traits Database is a freely accessible online
repository (https://doi.org/10.25365/phaidra.49; https://www.univie.ac.at/arctictraits) that fulfils these requests for
one important component of polar marine life, the Arctic benthic macroinvertebrates. It accounts for 1) obligate
traceability of information (every entry is linked to at least one source), 2) exchangeability among trait platforms
(use of most common download formats), 3) standardization (use of most common terminology and coding
scheme), and 4) user friendliness (granted by an intuitive web-interface and rapid and easy download options).
The combination of these aspects makes the Arctic Traits Database the currently most sophisticated online
accessible trait platform in (not only) marine ecology and a role-model for prospective databases of other marine
compartments or other (also non-marine) ecosystems. At present the database covers 20 traits (85 trait categories)
and holds altogether 8107 trait entries for 1211 macro- and megabenthic taxa. Thus, the Arctic Traits Database
will foster and facilitate trait-based approaches in polar regions in the future and increase our ecological
understanding of this rapidly changing system.

## 1 Introduction

The interest in trait-based approaches – i.e. such that consider the life history, morphological, physiological and
behavioral characteristics of species – in the marine realm has been growing tremendously in the last decades
(reviewed in Degen et al., 2018) (Fig. 1). Reasons for the increasing popularity of these approaches are that they
offer a variety of additional options to solely species-based methods: Traits can be analyzed across wide
geographical ranges and across species pools (Bernhardt-Römermann et al., 2011), they can be used to calculate a
variety of functional diversity indices (Schleuter et al., 2010), to estimate functional redundancy, or be used as
indicators of ecosystem functioning (Bremner et al., 2006). Given the rapid changes we observe in many marine
regions of the world, and especially in the Arctic Ocean (Wassmann et al., 2011), the potential to indicate
vulnerability to climate change and biodiversity loss, or to estimate climate change effects on ecosystem functions
is another inherent advantage of trait-based approaches.



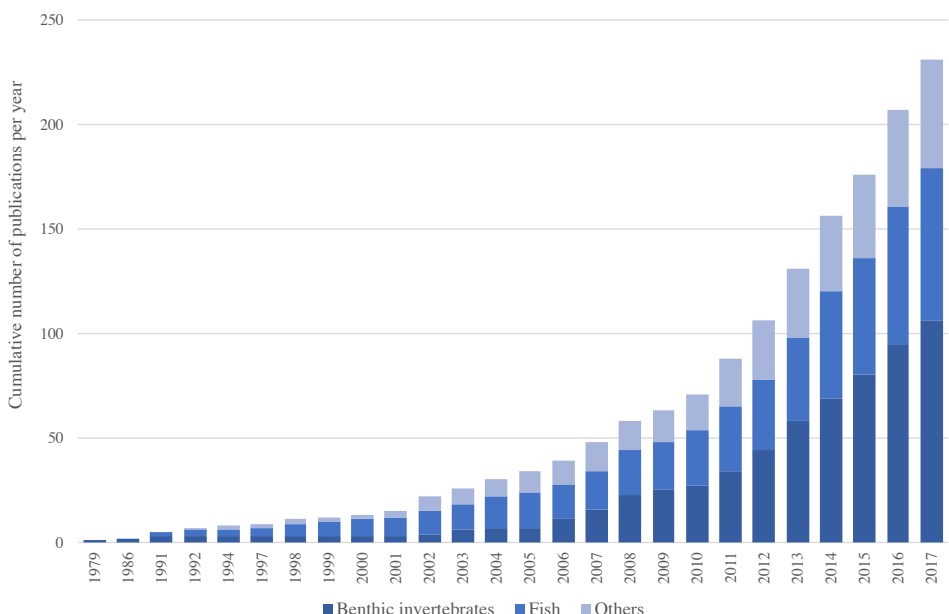

**Figure 1.** Cumulative number of marine trait-based studies based on the literature review of 233 studies from the marine realm by Degen et al. (2018).

Although the methodical diversity and complexity of trait-based approaches has broadened in the last years, the underlying data are always species traits. Trait information, however, is often not easy to find, and its collation requires a time and labor intensive survey of literature. This holds especially true for the polar regions, as ecological information for many polar marine taxa is still scarce, and only few publications supplement traceable resources of trait information (Kokarev et al., 2017). An additional obstacle is that existing trait repositories focus mainly on species from temperate regions. The increasing variability in terminology that surrounds traits is another challenge, and recent publications stress the importance of standardization in order to facilitate meta-analyses and comparison of results (Costello et al., 2015; Degen et al., 2018). Several online accessible trait databases specialize in specific taxonomic groups such as fish, polychaetes, or copepods, while others cover a wider part of the marine community (Table 1). The number of traits included and the form of access varies considerably among the different repositories. The database for marine copepods (Brun et al., 2017) contains 14 traits, whereas Fishbase (http://www.fishbase.org), polytraits (Faulwetter et al., 2014) and BIOTIC (http://www.marlin.ac.uk/biotic) contain more than 40 traits. Some repositories allow only for online browsing, while others enable different forms of download that range from spread sheets to different matrix formats (Table 1). No traits repository explicitly comprising polar species exists so far.

**Table 1.** List of marine trait databases or repositories. "Component" indicates the organism group targeted, "Access options" indicates in which forms the data can be accessed. Reference and web links are provided.

| Component | Access options | Publication, web links |
|---|---|---|
| Copepoda | Download of excel workbook via PANGAEA, traits provided as original values or binary code (0/1), references per trait provided. | (Brun et al., 2017) https://doi.pangaea.de/10.1594/ PANGAEA.862968 |



| Polychaeta | Download of full database or specified subsets in various formats (references and partly original quote and page number given), online via browsing the *Polychaetes Scratchpads* | Faulwetter et al. (2014) http://polytraits.lifewatchgreece.eu http://polychaetes.lifewatchgreece.eu |
|---|---|---|
| Benthos | Download of trait information in several matrix formats; as text and for certain traits as binary (0/1) code, also browsing online | Biological Traits Information Catalogue (BIOTIC) http://www.marlin.ac.uk/biotic |
| Fish | Browse online, programmatically via Application Programming Interface (API) and R package rfishbase | Froese, R. and D. Pauly. Editors. 2018. FishBase. www.fishbase.org, version (02/2018) |
| Benthos | Browse online | Marine Macrofauna Genus Trait Handbook, http://www.genustraithandbook.org.uk |
| Corals | Browse online, download as *.csv file | https://coraltraits.org/ |
| All marine | Browse online | Marine Species Traits, www.marinespecies.org/traits |
| All marine | Browse online | Sea Life Base, http://www.sealifebase.org |
| Fossil groups | Browse online | Neogene Marine Biota of Tropical America (NMiTA) http://eusmilia.geology.uiowa.edu |
| All biota | Browse online, programmatically via API | Encyclopedia of Life (EoL), http://www.eol.org |

With the here presented Arctic Traits Database we aim to bridge some of the above-mentioned issues for one important compartment of marine life: the Arctic macro- and megabenthic invertebrates. In order to fulfil the communities' demand for standardization and comparability only those traits and trait categories are included, that

are most frequently used in topical publications or which are already provided in freely accessible trait databases (Table 1). Regarding download options and traceability we follow the successful example given in Faulwetter et al. (2014) and provide download of trait data in different structured formats (atomized columns and DarwinCore). The use of these formats guarantees that the included trait information can be easily shared between trait repositories and that the content is fully exploitable both by humans and computers. Every trait code is backed up

by at least one reference, and where possible the original quote and page number are provided. In addition to above mentioned formats, for the first time trait information is made available also in a fuzzy-coded and ready-to-use matrix format, that can be directly incorporated into appropriate analysis software.

By providing the Arctic Traits Database to the community of benthic ecologists we aim to provide a sound basis for prospective trait-based approaches in polar regions which will in return aid our overall understanding of

these unique and rapidly changing ecosystems.

## 2 Data

### 2.1 Taxon data

The current taxa in the database are a subset of the dataset compiled in the frame of the "Arctic Traits Project" (Austrian Science Fund FWF, T801-B29), with focus on pan-Arctic benthic invertebrate macro- and megafauna.

This dataset comprises species lists from published studies of collaborators (Blanchard et al., 2013a, 2013b; Grebmeier et al., 2015), but also from so far unpublished sampling campaigns (e.g. field courses of the University Center in Svalbard, UNIS, 2007-2017). The regional coverage currently comprises the Chukchi Sea and the Svalbard area.

### 2.2 Trait data



Currently we consider 20 traits and 85 trait categories that reflect the morphology, life history, and the behavior of Arctic benthic invertebrates (Table 2). All traits are in categorical format, i.e. belonging to one out of up to six clearly defined trait categories (see Table 2). The four continuous traits included (body size, body weight, longevity, and depth distribution) are converted into categories, but the associated text information assures accessibility to users also in their original, numerical or continuous format.

The choice of which traits to include in the database is based on the following considerations: 1) trait information should be available for and applicable to all benthic taxa (Costello et al. 2015), 2) traits used in previous studies and databases should be favored to enable comparisons across studies (Degen et al. 2018), and 3) the traits should be usable across a wide geographical area (Bremner et al. 2006). Recent trait-based studies emphasize the importance of standardized traits and trait terminology to ensure that data can be integrated more
easily in the future (Costello et al 2015, Degen et al. 2018, Faulwetter et al. 2014). To meet these requirements of the scientific community, the Arctic Traits Database includes seven of the ten traits prioritized in Costello et al. (2015): "depth range", "substratum affinity", "mobility", "skeleton", "diet", "body size" and "reproduction" (Table 3). The remaining three traits emphasized in Costello et al. (2015) – taxonomic identity, environment, and geography – are not included. For taxonomic traits, every species in the database is deep linked to the World
register of Marine Species (WoRMS Editorial Board 2017; http://www.marinespecies.org/). For more detailed biogeographic information we refer users to the Global Biodiversity Information System (GBIF; http://www.gbif.org/) or the Ocean Biogeographic Information System (OBIS; http://www.iobis.org). We do include, however, the trait "zoogeography", which enables a differentiation between typical arctic and boreal or cosmopolitan taxa. Of the 20 traits used here, 17 are also identical to those used by the BIOTIC database (MarLIN
2006, Table 1), one of the most comprehensive databases on biological traits of marine organisms. Biotic also includes the trait "salinity". We cover salinity preferences within the trait "tolerance", which accounts also for temperature and pollution tolerance (see Table 3 for details). Traits we include in addition are "weight", "skeleton", and "mobility" (i.e. the relative degree of movement). Although physiological traits are of high interest in trait-based studies, we do not include them as they are not easily retrieved for many (arctic) benthic taxa (one of the
preconditions for inclusion in the database as stated above). In addition, physiological traits (e.g. growth rate, respiration rate, ingestion rate) depend on body mass and temperature (Brown et al., 2004), which can vary tremendously among Arctic regions, contradicting that the provided traits information should be usable across a wide geographical area.

**Table 2.** Trait terminology as used in the Arctic Traits Database, BIOTIC, Costello et al. 2015, and in "other" marine trait-based studies (i.e. studies reviewed in Degen et al. 2018, list non-exhaustive, see Appendix 1 of Degen et al. 2018 for total trait list and corresponding literature references). Be aware that the Arctic Traits Database and BIOTIC consider only benthic taxa, while Costello et al. (2015) and the studies summarized in "Other" cover all marine groups.

| Arctic Traits Database | BIOTIC | Costello et al. (2015) | Other |
|---|---|---|---|
| Body size | Body size | Body size | Body size/length/height, Largest radius, Biovolume, Coverage |
| Body weight | – | – | Body weight/mass, Biomass, Colony weight |
| Body form | Growth form | – | Body form, Body design, Body shape, Growth form, Growth type, Functional form group, Morphology |
| Fragility | Fragility | – | Fragility, Structural robustness, Shell strength |
| Skeleton | – | Skeleton | Skeletal composition/ thickness/material/density |
| Sociability | Sociability | – | Sociability, Schooling, Gregariousness, Social group size, Social behavior |
| Reproduction | Reproductive type | Reproduction | Reproduction, Reproduction type, Reproductive method/strategy/type/technique |



| Larval development | Developmental mechanism | – | Larval development, Larvae type, Larval feeding, Larval development location, Developmental mode/type/mechanism/technique |
|---|---|---|---|
| Longevity | Life span | – | Longevity, Age, Life span, Maturity, Life duration, Generation time |
| Environmental position | Environmental position | – | Environment, Environmental position, Habitat, Vertical distribution, Sediment position, Living position, Life zone |
| Living habit | Living habit | – | Living habit, Habit, Life habit, Life form, Habitat, Living mode, Habitat structure |
| Mobility | – | Mobility | Mobility, Relative mobility, Degree of mobility, Mobility within sediment |
| Adult movement | Mobility/Movement | – | Adult movement, Mobility, Movement method/type, Locomotion |
| Feeding habit | Feeding habit | – | Feeding habit/behavior/method/type/apparatus, Resource capture method, Trophic mode, Oral gape position/height/surface, Protrusion |
| Trophic level | Typical food types | Diet | Trophic level, Diet, Food type, Trophic group, Dietary group |
| Bioturbation | Bioturbation | – | Bioturbation mode/type/potential, Sediment movement/reworking/transport, Direction of sediment transport, Reworking mode, Fecal deposition, Irrigation |
| Tolerance | Salinity | – | Tolerance, Tolerance limits, Salinity tolerance, Survival salinity/temperature, Temperature optimum, Thermal affinity, Hypoxia tolerance, Tolerance to pollutants, Ecological group, Resilience, Condition index |
| Zoogeography | Biogeographic range | – | Biogeography, Geographical range/distribution, Range size, Native region, Median latitude |
| Depth range | Biological zone | Depth range | Depth range/regime, Diving depth |
| Substratum affinity | Substratum affinity | Substratum affinity | Substratum affinity, Habitat, Habitat preference/type/specifity/complexity, Preferred substrate, Substrate type, Living location |

One common approach to use traits is as indicators of ecosystem functions (effect traits) or of changes in the environment (response traits) (Degen et al. 2018). An overview of how each of the 20 traits that are currently included in the database may relate to ecosystem functions or respond to environmental changes or pressures is given in Table 3.

**Table 3.** Detailed information on the 20 biological traits currently included in the Arctic Traits Database, clustered into morphology traits (6), life history traits (3), and behavioral traits (11). For every trait and its categories, the definition as used in the Arctic Traits Database is given. Abbreviations of each category are given (e.g. S1, S2) as these are used in files downloaded from the website. The relation of the respective trait to benthic ecosystem functions or responses and the underlying literature sources are displayed as well.

**MORPHOLOGY**

| Body Size | | |
|---|---|---|
| Definition | Maximum body size as adult given in mm, as individual or colony and excluding appendages. Can be height in rather upright animals (e.g. corals), body width or diameter in rather round animals (e.g. crabs), or body length in elongated animals (e.g. worms). | |
| Categories | S1 small | < 10 mm |
| | S2 small-medium | 10-50 mm |
| | S3 medium | 50-100 mm |
| | S4 medium-large | 100-300 mm |
| | S5 large | > 300 mm |
| Function | Size has a direct effect on productivity, the amount of habitat structuring and facilitation, and is important for the amount of oxygen and nutrient flux across the sediment-water interface. It correlates with food web structure, trophic levels, and energy flow in ecosystems. | |
| Detail | Smaller animals are faster growing, usually show a higher productivity and are less affected by trawling as they are more likely to fit through the net of trawling gear, thus often replacing larger slow-growing fauna in trawl-impacted areas. A clear majority of small-bodied species may be indicative for environments with high instability or be the result of environmental or anthropogenic disturbances. Larger taxa usually show a lower productivity but higher carbon | |




| | fixation and have a higher effect on fluxes of nutrients, energy and matter. They usually grow slower, reproduce later, and are more affected by trawling and other disturbances. |
|---|---|
| References | Bolam and Eggleton, 2014; Bremner, 2008; Costello et al., 2015; Emmerson, 2012; Micheli and Halpern, 2005; Norkko et al., 2013; van der Linden et al., 2016 |

**Body weight**

| | | | |
|---|---|---|---|
| Definition | The wet weight (WW) of an (adult) organism given in gram (g), including shell or skeleton. | | |
| Categories | W1 | low | < 0.1 g |
| | W2 | low-medium | 0.1-1.0 g |
| | W3 | medium | 1.0-10 g |
| | W4 | medium-high | 10-100 g |
| | W5 | high | > 100 g |
| Function | Weight affects metabolic rate, energy demand and uptake rate. Animals with lighter weight have usually a faster life cycle and higher productivity, while heavier animals usually have a slower life cycle and productivity, but higher carbon fixation. | | |
| Remark | Closely linked to body size. | | |
| References | Bolam and Eggleton, 2014; Bremner, 2008; Costello et al., 2015; Norkko et al., 2013 | | |

**Body form**

| | | | |
|---|---|---|---|
| Definition | The external characteristic of an organism. | | |
| Categories | BF1 | globulose | Round or oval (e.g. sea urchin, sponge, some bivalves) |
| | BF2 | vermiform | Wormlike |
| | BF3 | dorso-ventral compressed | Species that are flat, or encrusting (e.g. starfish, sponge) |
| | BF4 | laterally compressed | Thin (e.g. isopods, amphipods, some bivalves) |
| | BF5 | upright | E.g. coral, basket star, sponge |
| Function | The body form can be indicative for the ecological role of species in an ecosystem (e.g. if it is habitat-forming), and for its vulnerability to mechanical disturbances (e.g. bottom trawling). Species with an upright body form will be more affected than vermiform or flat ones. Sets restrictions to habitat use and migration capability. Vermiform taxa can be a proxy for litter quality/decomposition. | | |
| Remark | Often simply a proxy of taxonomy (e.g. vermiform > polychaetes, laterally compressed > amphipods). | | |
| References | Beauchard et al., 2017; Bolam and Eggleton, 2014; Costello et al., 2015; Törnroos and Bonsdorff, 2012; Wiedmann et al., 2014 | | |

**Fragility**

| | | | |
|---|---|---|---|
| Definition | The degree to which an organism can withstand physical impact. | | |
| | F1 | fragile | Likely to crush, break, or crack as a result of physical impact (e.g. brittle star, soft worms, smaller crustaceans, mollusks with thin shells) |
| | F2 | intermediate | Liable to suffer minor damage, chips or cracks as result of physical impacts (e.g. mollusks with thicker shells, animals with harder cuticle like some echinoderms) |
| | F3 | robust | Unlikely to be damaged as a result of physical impacts, e.g. hard or tough enough to withstand impact, or leathery or wiry enough to resist impact (e.g. starfish, sponges, tunicates) |
| Function | Determines sensitivity to physical disturbance (e.g. bottom trawling) and to predatory aggression. Softer/fragile bodies are stronger affected by trawling. Indicative for prey accessibility and ease of ingestion. | | |
| References | Beauchard et al., 2017; Bolam and Eggleton, 2014; Weigel et al., 2016 | | |

**Skeleton**

| | | | |
|---|---|---|---|
| Definition | Presence and type of supporting structures in the animal body. | | |
| Categories | SK1 | calcareous | Skeleton material aragonite or calcite (e.g. bivalves) |
| | SK2 | siliceous | Skeleton material silicate (e.g. siliceous sponges) |
| | SK3 | chitinous | Skeleton material chitin (e.g. arthropods) |
| | SK4 | cuticle | No skeleton but a protective structure like a cuticle (e.g. sea-squirts) |
| | SK5 | none | No form of protective structure (e.g. sea slugs) |
| Function | Indicates vulnerability (trawling, ocean acidification), resistance to predation (proxy of palatability), and ecosystem engineering (provision of habitat, increased heterogeneity). Large calcifying taxa contribute most to inorganic carbon sequestration. | | |
| References | Costello et al., 2015; Frid and Caswell, 2016, 2015; Spitz et al., 2014 | | |

**Sociability**



| Definition | The degree to which species aggregate. | | |
|---|---|---|---|
| Categories | SO1 | solitary | Single individual |
| | SO2 | gregarious | Single individuals forming groups; growing in clusters (e.g. barnacles) |
| | SO3 | colonial | Living in permanent colonies (e.g. stony corals, Bryozoa, Synascidia) |
| Function | Determines sensitivity to physical disturbance (e.g. bottom trawling) and can indicate if a species can increase habitat heterogeneity or is habitat forming. If yes, then it affects habitat creation, nursery, refuge, facilitation, and sediment oxygenation. | | |
| References | Beauchard et al., 2017; Costello et al., 2015 | | |

## LIFE HISTORY TRAITS

| Reproduction | | | |
|---|---|---|---|
| Definition | The way species reproduce, here including information about where fertilization occurs and whether propagules are released or not. | | |
| Categories | R1 | asexual | Budding and fission (e.g. sponges, cnidarians) |
| | R2 | sexual – external | Fertilization external, eggs & sperm deposited on substrate or released into water (broadcast spawners) (e.g. echinoderms, cnidarians) |
| | R3 | sexual – internal | Fertilization internal, but no brooding, eggs deposited on substrate, indirect or direct development (e.g. gastropods) |
| | R4 | sexual – brooding | Fertilization internal or external, Eggs or larvae are brooded, indirect or direct development (e.g. amphipods, isopods, echinoderms) |
| Function | Indicates the ability of a species to disperse, become invasive, or recover from a population decline. Can indicate if carbon is transported from the benthic to the pelagic realm or stays locally bound. Animals without a planktonic stage that perform brooding and parental care might have a higher tolerance against some forms of stress (e.g. ocean acidification), but may be higher vulnerable to local disturbances (biotic or abiotic). | | |
| References | Bremner, 2008; Costello et al., 2015; Lucey et al., 2015 | | |

| Larval Development | | | |
|---|---|---|---|
| Definition | Larval development and feeding type. | | |
| Categories | LD1 | pelagic/planktotrophic | High fecundity, larvae feed and grow in water column, generally pelagic for several weeks (e.g. echinoderms, bivalves) |
| | LD2 | pelagic/lecitotrophic | Medium fecundity, larvae with yolk sac, pelagic for short periods (e.g. tunicates) |
| | LD3 | benthic/direct | Larvae have benthic or direct development (no larval stage, eggs develop into miniature adults) |
| Function | Ability of a species to disperse, become invasive, or recover from a population decline. Indicator for long-term sensitivity (ability to recolonize disturbed areas). Planktonic stages indicate productivity and elemental transport from benthos to pelagos. | | |
| References | Bolam and Eggleton, 2014; Cardeccia et al., 2018; Törnroos and Bonsdorff, 2012 | | |

| Longevity | | | |
|---|---|---|---|
| Definition | The maximum reported life span of the adult stage in years. | | |
| Categories | A1 | short | <2 years |
| | A2 | medium | 2-5 years |
| | A3 | medium-long | 5-20 years |
| | A4 | long | >20 years |
| Function | Long lived animals are more susceptible to disturbance and need longer to recover (while short-lived species can recover fast and may increase in richness and abundance as disturbance increases). An indicator for population stability over time, carbon fixation, productivity. | | |
| Detail | Indicates the relative investment of energy in somatic rather than reproductive growth and the relative age of sexual maturity. A proxy for relative r- and k-strategy. | | |
| References | Bolam and Eggleton, 2014; Bremner, 2008; Cain et al., 2014; Costello et al., 2015 | | |

## BEHAVIORAL TRAITS

| Environmental Position | | | |
|---|---|---|---|
| Definition | The position of the animal relative to the sediment. | | |
| Category | EP1 | infauna | Lives in the sediment |



| | EP2 | epibenthic | Lives on the surface of the seabed |
|---|---|---|---|
| | EP3 | hyper-benthic | Living in the water column, but (primarily/occasionally) feeds on the bottom; bentho-pelagic |
| Function | Affects carbon fixation and transport within the sediment, between aerobic and anaerobic layers, or from pelagos to benthos. Can indicate facilitation (e.g. for microbial communities in the sediment) and sensitivity to perturbation (e.g. bottom trawling, infauna less affected than epifauna, hyper-benthic taxa might be able to escape). Endobenthic life style effects the sediment biogeochemistry. Epibenthic and shallow sediment-dwelling taxa are more vulnerable to predation. Hyper-benthic taxa are involved in transport of carbon from benthos to pelagos. | | |
| References | Bolam et al., 2014; Bremner et al., 2008; Frid and Caswell, 2016; Törnroos & Bonsdorff, 2012 | | |

| **Living Habit** | | | |
|---|---|---|---|
| Definition | The mode of living, ranging from free over tube or burrow dwelling to permanently attached. | | |
| Categories | LH1 | free living | Not limited to any restrictive structure at any time. Able to move freely within and/or on the sediments |
| | LH2 | crevice dwelling | Adults are typically cryptic, inhabiting spaces made available by coarse/rock substrate and/or biogenic species or algal holdfasts |
| | LH3 | tube dwelling | Tube may be lined with sand, mucus or calcium carbonate, tube can also be in a burrow |
| | LH4 | burrowing | Species inhabiting permanent or temporary burrows in the sediment, or are just burrowing in the sediment |
| | LH5 | epi/endo zoic/phytic | Living on or in other organisms |
| | LH6 | attached | Adherent to a substratum |
| Function | Attached species are more vulnerable to predation and perturbations (e.g. bottom trawling). Burrowing, crevice and tube dwelling taxa affect sediment biogeochemistry, carbon transport, elemental cycling, and are less affected by strong hydrodynamic disturbance, anoxic conditions and water pollution. Tube building can add to local storage of chemicals and waste materials. Microbial processes are facilitated and microbial biomass promoted by deep-dwelling fauna. Burrowing and irrigation generally facilitates life of associates. Burrowing or attached living habit can be related to habitat creation and facilitation. | | |
| References | Aller, 1983; Bolam and Eggleton, 2014; Bremner, 2008; Bremner et al., 2006; Costello et al., 2015; Törnroos and Bonsdorff, 2012; van der Linden et al., 2016 | | |

| **Mobility** | | | |
|---|---|---|---|
| Definition | Degree or intensity of movement. | | |
| Categories | MO1 | none | No movement as adult (sponge, coral) |
| | MO2 | low | Slow movement (e.g. anemones, snails) |
| | MO3 | medium | Medium movement (e.g. starfish, brittle stars) |
| | MO4 | high | High movement, swimmer or fast crawler (e.g. amphipods, shrimp) |
| Function | Slowly or non-moving species are more vulnerable to predation, perturbations and decrease in food input, while mobile taxa are more flexible and may evade trawl gear or predators. High percentage of non-moving organisms can indicate high amount of food, while high percentage of highly mobile taxa may indicate food patchiness or scarcity. Indicative for dispersal potential and ability to recolonize. | | |
| References | Costello et al., 2015; Micheli and Halpern, 2005; Tyler et al., 2012 | | |

| **Adult movement** | | | |
|---|---|---|---|
| Definition | Type of movement as an adult. | | |
| Categories | MV1 | sessile/none | No movement as adult (sponge, coral) |
| | MV2 | burrower | Movement in the sediment (e.g. annelids, echinoderms, crustaceans, bivalves) |
| | MV3 | crawler | An organism that moves along on the substratum via movements of its legs, appendages or muscles (e.g. crabs, snails) |
| | MV4 | swimmer (facultative) | Movement above the sediment (e.g. amphipods) |
| Function | Indicates the dispersal and recolonization potential, and the invasiveness of an organism. Related to nutrient cycling (burrowing taxa contribute most to nutrient cycling and regeneration, burrows increase the total sediment surface area available for exchange with the water column), carbon deposition (sessile calcifying taxa), facilitation of microbial and other fauna (either via burrowing or via constructing biogenic habitats), and habitat stability. Swimmers may escape predators and trawling gear. | | |
| Remark | Closely linked to the trait mobility. | | |
| References | Aller, 1983; Bremner, 2008; Bremner et al., 2006; Costello et al., 2015; Frid and Caswell, 2016 | | |

| **Feeding Habit** | | | |
|---|---|---|---|



| Definition | The mode of food uptake. | | |
|---|---|---|---|
| Categories | FH1 | surface deposit feeder | Active removal of detrital material from the sediment surface. Includes species which scrape and/or graze algal matter from surfaces |
| | FH2 | subsurface deposit feeder | Removal of detrital material from within the sediment matrix (e.g. Echinocardium) |
| | FH3 | filter/suspension feeder | Sponge, coral, hydrozoa, bivalves |
| | FH4 | opportunist/scavenger | An organism that can use different types of food sources/an organism that feeds on dead organic material (e.g. crabs, whelks) |
| | FH5 | predator | An organism that feeds by preying on other organisms (e.g. starfish) |
| | FH6 | parasite/commensal | An organism that lives in or on another living organism (the host), from which it obtains food and other requirements |
| Function | Can indicate hydrodynamic conditions (suspension feeders in turbulent, deposit feeders in calmer water), carbon transport between pelagos and benthos (suspension feeders) and backwards (predators), and vulnerability (e.g. surface deposit feeders and suspension feeders are more sensitive to trawling). Impacts resource utilization and facilitation (e.g. deposit feeders facilitate microbes that further decompose organic carbon). Effects the depth of oxygen and detritus penetration and can enhance organic matter decomposition and nutrient recycling/regeneration. Control of other species in the assemblage. | | |
| References | Bremner, 2008; Bremner et al., 2006; Dolbeth et al., 2009; Frid et al., 2008; Kröncke, 1994; Oug et al., 2012; Rosenberg, 1995; Tyler et al., 2012; van der Linden et al., 2016 | | |

| **Trophic Level** | | | |
|---|---|---|---|
| Definition | Rank of an animal according to how many steps it is above the primary producers at the base of the food web. | | |
| Categories | TL1 | 1 | Primary producer |
| | TL2 | 2 | Primary consumers – Herbivore / Deposit Feeder /Suspension Feeder |
| | TL3 | 3 | Secondary consumers – Carnivore |
| | TL4 | 4 | Tertiary consumers |
| | TL5 | 5 | Quaternary consumers – Apex predator |
| Function | Determines the role of an organism in energy transfer within the food web. Control of other species abundance in the assemblage. | | |
| References | Costello et al., 2015; Micheli and Halpern, 2005; Renaud et al., 2011 | | |

| **Bioturbation** | | | |
|---|---|---|---|
| Definition | Biogenic modification of sediments through living, movement and feeding habits of organisms. | | |
| Categories | B1 | diffusive mixing | Surficial movement of sediment and/or particles, resulting from movement or feeding activities on the surface |
| | B2 | surface deposition | Deposition of particles at the sediment surface resulting from e.g. defecation or egestion (pseudofaeces) by, for example, surface deposit feeding organisms (e.g. holothuroids, bivalves, tubiculous polychaetes) |
| | B3 | conveyor belt transport (upward) | Translocation of sediment and/or particulates from depth within the sediment to the surface during subsurface deposit feeding or burrow excavation |
| | B4 | downward (reverse) conveyor | The subduction of particles from the surface to some depth by feeding or defecation |
| | B5 | none | No bioturbation (e.g. sessile animals on hoard bottom) |
| Function | Impacts sediment biogeochemistry (oxygen, pH and redox gradients, elemental carbon), organic matter regeneration, nutrient cycling, sediment granulometry, pollutant release, microbial composition, abundance and diversity and in general provision and maintenance of habitats for other organisms. | | |
| References | Chen et al., 2017; Frid et al., 2008; Gogina et al., 2017; Lacoste et al., 2018; Mermillod-Blondin, 2011; Pearson, 2001; Queirós et al., 2013; Solan et al., 2012 | | |

| **Tolerance** | | | |
|---|---|---|---|
| Definition | Degree to which a species reacts to changes in its environment. | | |
| Categories | T1 | low | Species reacts sensitive to changes in the environment like organic enrichment, pollution, temperature or salinity changes; AMBI group I |
| | T2 | intermediate | Species react indifferent or no information available; AMBI group II |



| | T3 | high | Species tolerates organic enrichments, pollution, temperature or salinity changes; AMBI groups III-IV |
|---|---|---|---|
| Function | Indicates vulnerability or resistance/resilience of a species towards pollution or climate change induced changes in water biogeochemistry. | | |
| References | Borja and Franco, 2000; Gusmao, 2017; Marchini et al., 2008; Piló et al., 2016 | | |

| **Zoogeography** | | | |
|---|---|---|---|
| Definition | Distribution of a species (arctic, arctic-boreal, boreal, cosmopolite) | | |
| Categories | Z1 | arctic | Arctic distribution |
| | Z2 | arctic-boreal | Arctic-boreal distribution |
| | Z3 | boreal | Boreal distribution |
| | Z4 | cosmopolite | Cosmopolite distribution |
| Function | Indicates vulnerability (arctic species may be more vulnerable to changes than species with an arctic-boreal or cosmopolite distribution) or potential of a species to become invasive. | | |
| References | Fetzer, 2005; Fetzer and Arntz, 2008; Węsławski et al., 2003 | | |

| **Depth range** | | | |
|---|---|---|---|
| Definition | Species distribution related to water depth. | | |
| Categories | DR1 | shallow | 0-20 m |
| | DR2 | shelf | 20-200 m (some shelves can extend to 500 m) |
| | DR3 | shelf-slope | 200-1000 m (sometimes the slope starts deeper, e.g. 500-) |
| | DR4 | slope-basin | > 1000 m |
| Function | Can be used – along substratum affinity – for habitat classification. Can depict depth distribution of other traits. | | |
| Detail | Shallow water and shelf taxa face a higher exposure to predation of marine mammals and to physical disturbance such as iceberg scouring and to coastal processes and pollution. | | |
| References | Costello et al., 2015; Gutt, 2001 | | |

| **Substratum Affinity** | | | |
|---|---|---|---|
| Definition | Type of substratum that organisms (preferential) live on. | | |
| Categories | SA1 | soft | Soft substrata, sand or mud |
| | SA2 | hard | Hard substrata, rock, gravel |
| | SA3 | biological | Epizoic or epiphytic life style |
| | SA4 | none | Species is hyper/supra benthic and has no affinity for a certain substrate, but it might prefer one for hunting/scavenging (this category should not occur too often, as we work with benthos) |
| Function | Can be used – along depth range – for habitat classification. Can depict potential substrate specificity of other traits. | | |
| References | Costello et al., 2015 | | |

### 2.3 Sources of trait information

Sources of trait information are research papers, books, databases and online repositories (Table 1), but also grey literature such as cruise reports. Trait information can also result from onsite measurements, personal observations, or be transmitted via communication with experts for a specific taxonomic group. In any case, the source is indicated as precise as possible, for published literature with complete reference and DOI (if available), in case of expert communication the name and contact details of the respective expert are given. Wherever possible the original quote from literature and page numbers are given to ensure the traceability of the provided trait information. Although literature sources targeting the Arctic are used preferably (and for exclusively Arctic species are the only option) we do not restrict source information for arctic-boreal or cosmopolite taxa to stem from Arctic regions. This bears the risk that the assigned trait information is not accurate, as polar taxa might differ in their expression of certain traits from their relatives at lower latitudes (Degen et al. 2018). However, this is an issue for now not resolved, as trait information from the high latitudes is often scarce, and we recommend the user to consider the source of trait information when interpreting results.

### 2.4 Fuzzy coding of traits





The fuzzy coding procedure indicates to which extent a taxon exhibits each trait category (Chevenet et al., 1994). This method has the advantage that it enables us to analyze diverse kinds of biological information derived from a variety of sources (as those included in the Arctic Traits Database, see Sect. 2.3), and that also intermediate scenarios (i.e. when a taxon does not clearly fall into one category or the other) can be accounted for (Chevenet et al. 1994). We use the 0–3 coding scheme (details in Table 4 below) as it is the most widely used (which facilitates comparisons and exchange of trait information) and provides a compromise between binary codes and many not clearly delineated graduations (Degen et al. 2018).

**Table 4.** Explanation of fuzzy codes as used in the Arctic Traits Database.

| Fuzzy code | Explanation |
|---|---|
| 3 | Taxon has total and exclusive affinity for a certain trait category, all other categories do not apply and must be coded with "0". |
| 2 | Taxon has a high affinity for a certain trait category, but other categories can occur with equal (2) or lower (1) affinity. |
| 1 | Taxon has a low affinity for a certain trait category. |
| 0 | Taxon has no affinity for a certain trait category. |

**Table 5.** Two coding examples for the trait "Feeding habit" which has six trait categories (FH1 – FH6, see also Table 3). Species 1 is a surface deposit feeder, but can switch facultative to suspension feeding, while species 2 is an exclusive suspension feeder.

| Feeding habit | Abbreviation | Species 1 | Species 2 |
|---|---|---|---|
| Surface deposit feeder | FH1 | 2 | 0 |
| Subsurface deposit feeder | FH2 | 0 | 0 |
| Filter/suspension feeder | FH3 | 1 | 3 |
| Opportunist/scavenger | FH4 | 0 | 0 |
| Predator | FH5 | 0 | 0 |
| Parasite/commensal | FH6 | 0 | 0 |

**Table 6.** This is how the above example would appear in the matrix downloaded from the Arctic Traits Database. In the download matrix format species are rows, trait categories are columns, and the fuzzy codes are the values. Due to the database structure zero codes ("0") are only displayed when they are backed up by a specific reference (e.g. for the trait category LH3/tube dwelling: "No species within the family Polynoidae is tubiculous").

|  | FH1 | FH2 | FH3 | FH4 | FH5 | FH6 |
|---|---|---|---|---|---|---|
| Species 1 | 2 |  | 1 |  |  |  |
| Species 2 |  |  | 3 |  |  |  |

While the coding might for some traits and taxa be pretty straight forward, in some cases a decision might be drawn not so easily. As one of the clearer cases, we point out the coding of the trait "body size" for the star fish *Crossaster papposus*. A literature reference states that the body size can range "Up to 340 mm in diameter" (Hayward and Ryland, 2012, p. 668). This size fits into the category "large" (S5, > 300 mm), thus the taxon is coded "3" for this size class, and "0" for all other categories (S1 – S4). The trait "mobility" is trickier. A literature reference (Himmelman and Dutil, 1991), p. 68) states the following: "*Crossaster papposus* and *Solaster endeca* are highly mobile; large individuals can cover distances of more than 5 meters in 12 hours". Here we have to keep in mind that the particular reference frame in this publication are subtidal sea stars in the northern Gulf of St. Lawrence (West Atlantic). The reference of the Arctic Traits Database however are all benthic invertebrates, and the trait category "high mobility" is defined here for taxa which are "swimmers or fast crawlers", such as some





amphipods and shrimp (see Table 2). Accordingly, the correct coding for *C. paposus* in the reference system of the Arctic Traits Database is the category "medium" mobility (MO3). Users of the Arctic Traits Database should bear this reference system in mind when downloading only the fuzzy coded trait data and aiming to apply it to another reference system. But as the detailed literature quote that lead to the coding of a trait is always provided

(see Sect. 2.3), the trait information can easily be adjusted by the user.

There will always be a certain degree subjectivity related to the fuzzy coding procedure. To find out how strong the coding might differ among scientists a small experiment at the Arctic Traits Workshop in Vienna (December 2016) was performed (Degen et al. 2018). Participants coded 27 trait categories of three common Arctic benthic species, and found the final trait matrices to be to 83% identical. We are confident that the

sophisticated structure of the Arctic Traits Database (see Sect. 3) and the provided information and instructions will support a more consistent coding of benthic traits in the future.

### 3 Database

In order to collect trait information and to disseminate it among users, a web-based database was created. The database features a public interface (Sect. 3.1) and an entry interface that is accessible only for registered

collaborators (Supplement). The public interface (Fig. 2, a) allows to browse the traits and references online ("Data per taxon" in the top menu bar), to view background information ("About" and "Trait definitions") and to download either the entire species, trait and literature information or specified subsets in several formats ("Download data") (see Sect. 3.1). Registered collaborators – i.e. those users that actively contribute trait information to the Arctic Traits Database – can access the interactive part of the database via the log in button on

the public page (Fig. 2a). This access offers additional options (Fig. 2b): browsing the existing information also per traits ("Traits" in the top menu bar), uploading new taxa, trait and source information, or adding trait information, references and comments to already existing taxa in the database ("Taxa"). As several users can work on the same taxa, a flagging system is used to highlight and discuss potentially conflicting sources and opinions. The "References", "Statistics", and "Tools" sections are equally accessible only for registered users (Fig. 2, b;

Supplement). Every scientist working in the field of Arctic benthic ecology aiming to share trait information can become a registered user by getting in touch with the editor and retrieving a user login. Credit to the registered collaborators is given in the "About" section on the public site and also on taxon pages after each trait entry they conduct. A detailed manual for registered users is provided in the supplementary material to this publication (Supplement), or can alternatively be accessed via the public web interface ("About"). Collaborators who want to

share trait information without registering to the database can alternatively be provided with an upload template (.xls).



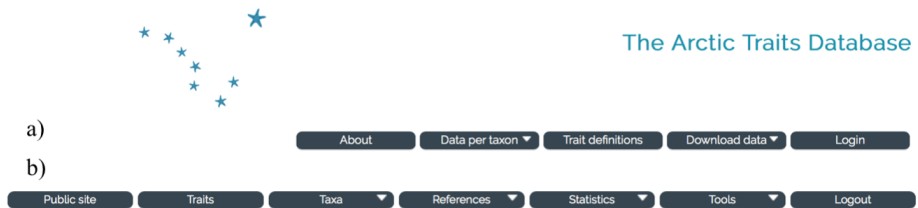

**Figure 2.** Screenshots of the start page of the Arctic Traits Database. Toolbar of the public page with Login button for the registered user (a), and toolbar in the area for registered users (b).

### 3.1 Public access and download options

The public access enables to browse the database online and to download the complete set of data as well as the bibliography, or specified subsets. Taxon traits can be visually inspected online via the "Data per taxon" button from the top menu bar and "Browse taxa". Taxa can be browsed and selected via the taxonomic tree, as indicated

for the asteroid *Crossaster papposus* in Fig. 3.

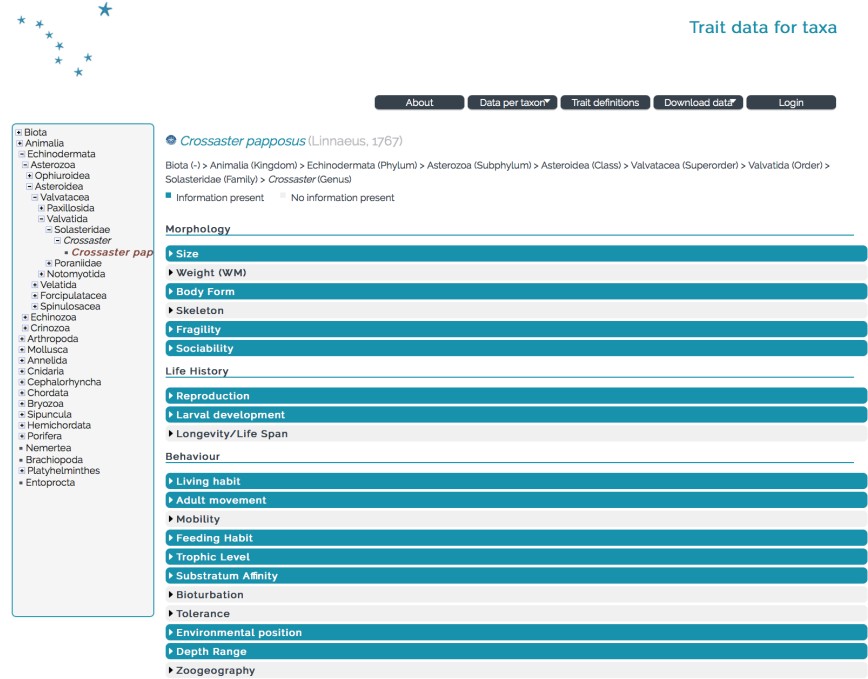

**Figure 3.** Screenshot of the taxon page of the asteroid *Crossaster papposus* selected from the classification tree on the left.

The completeness of trait information can be inspected via "Data completeness" (Fig. 4), equally accessible via "Data per taxon" on the top menu bar.  This option shows an alphabetic list of all taxa in the database for which



trait information is available. The bar on the right side indicates the information coverage for each taxon and trait, blue color indicates that trait information is present.

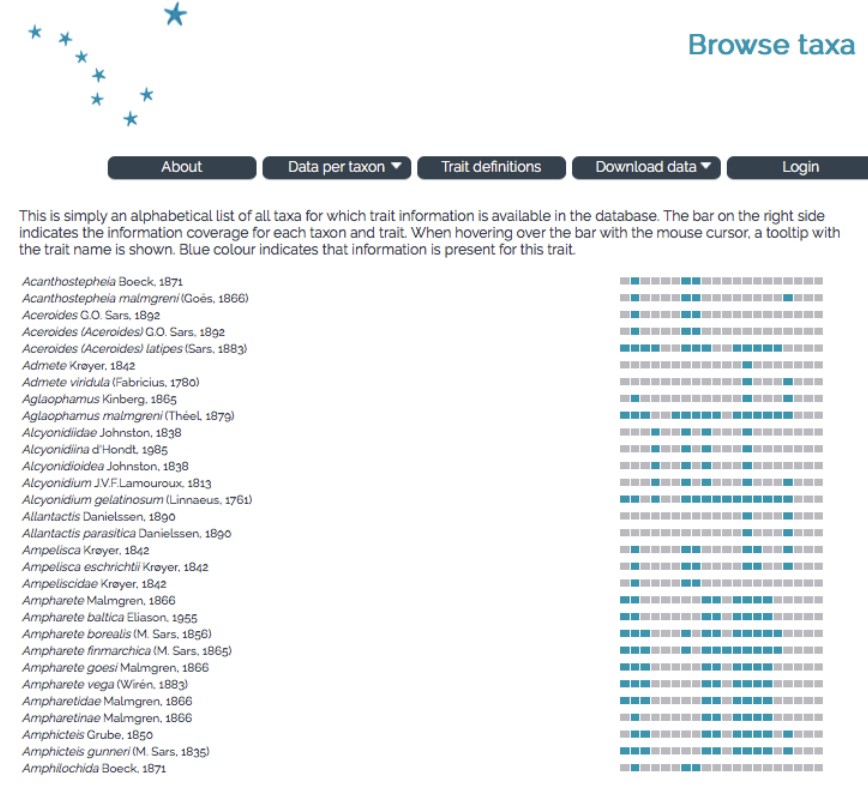

**Figure 4.** Screenshot of data completeness.

The download section can be accessed via the "Download data" button on the top menu bar (Fig. 2, a; Fig. 3; Fig. 4). Download is enabled in three different computer readable formats: 1) as data in columns (*.csv) (Table 7), 2) in DarwinCore format (Table 8), and 3) as fuzzy coded trait matrix which some users might prefer (see Sect. 2.4

and Fig. 5). Also, the entire bibliography is available for download. Before the download commences the user is asked whether to download a) all data in the database, b) only data for an uploaded list of taxon names, c) only data for an uploaded list of AphiaIDs, or d) only the data selected from a classification tree. In the last option, entire phyla or sub-groups can be easily selected from the tree. By default, all 20 traits are exported, but if the user is interested only in one or a few specific traits, the option to select these from the total list of 20 traits is available.

As the fuzzy coded trait matrix (download option 3) contains only the fuzzy codes per trait category but no literature sources, we recommend to also download the "Data in columns" (download option 1) for the same taxa, where the detailed source per species and trait category is included. Details on the structure of the first two download options are given below in Table 7 and Table 8. A clipping from a downloaded fuzzy coded trait matrix




is shown in Fig. 5. The database can also be accessed programmatically via a REST API (documented at

https://www.univie.ac.at/arctictraits/download-api).

**Table 7.** List of fields returned by the Arctic Traits Database when "Data as columns" (*.csv) is chosen as an export option from the download section.

| Column label | Column description |
| --- | --- |
| Taxon | The taxon for which the information was recorded. |
| Author | The author and year of the *Taxon* for which the information was recorded. |
| Valid taxon | Currently accepted name of the *Taxon* (as stored in the Arctic Traits Database - information might not be up to date with the WoRMS or the latest taxonomic literature in some cases). Users should check all taxa against WoRMS before use. If *Taxon* is currently accepted, this field contains the same value as *Taxon*). |
| Valid author | Currently accepted name of the *Author* (as stored in the Arctic Traits Database - information might not be up to date with the WoRMS or the latest taxonomic literature in some cases). Users should check all taxa against WoRMS before use. If *Taxon* is currently accepted, this field contains the same value as *Author*. |
| Source of synonymy | Literature reference for synonymy of taxon (if present). |
| Parent taxon | The *Taxon*'s direct parent in the taxonomic classification (as stored in the Arctic Traits Database). |
| Trait | The biological trait for which information is available (e.g. "Feeding habit"). |
| Category | The sub-category of the *Trait* for which information is available (e.g. "Predator"). |
| Category abbreviation | An abbreviated version of the often verbose trait category - useful as a label in further analyses of the data (e.g. "FH(6)"). |
| Traitvalue | Describes the affinity of the *Taxon* to the *Category*. Values range from 0–3: "0"= no affinity for a certain trait category; "1"= low affinity for a certain trait category; "2"= high affinity for a certain trait category, but other categories can occur with equal (2) or lower (1) affinity; "3"= total and exclusive affinity for a certain trait category. |
| Reference | Literature reference leading to the assignment of the *Traitvalue* to the *Category* for the *Taxon*. |
| DOI | Digital Object Identifier (where available) of the *Reference*. |
| Value creator | Person who assigned the *Traitvalue* to the *Category* for the *Taxon*, supported by a *Reference*. |
| Value creation date | Date and time when the above information was entered into the database. |
| Text Excerpt | A quotation of the original text passage from the literature source that led to the assignment of assignment of the *Category/Traitvalue* to the *Taxon*. Empty if information has not been recorded yet. |
| Text Excerpt creator | Person who entered the *Text excerpt*. Only present if *Text Excerpt* is present. |
| Text Excerpt creation date | Date and time when the *Text Excerpt* was entered into the database. Only present if *Text Excerpt* is present. |

**Table 8.** List of fields returned by the Arctic Traits Database when "Darwin Core" is chosen as an export option from the download section. DarwinCore does not provide the same granularity as the "Data as columns" format. The output file consequently contains fewer details.

| Column label | Column description |
| --- | --- |
| scientificName | The taxon for which the information was recorded |
| scientificNameAuthorship | The author and year of the taxon for which the information was recorded |
| acceptedNameUsage | Currently accepted name and authorship of the *scientificName* (as stored in the *arctictraits* database – information might not be up to date with the latest taxonomic literature in some cases.) |
| Taxonomic Status | The status of the use of the *scientificName* (e.g. objective synonym, subjective synonym) as stored in the *arctictraits* database. Empty if *scientificName* is the currently accepted name. |
| MeasurementOrFact | Trait name and trait category, separated by a colon (e.g. Size:small) |
| measurementValue | Value from 0–3, describing the affinity of the taxon to a trait category. Coding of values as described in Table 7 "Traitvalue". |
| dcterms:bibliographicCitation | Full literature reference (including Digital Object Identifier (DOI) where present) supporting the trait information for the current taxon. |
| measurementRemarks | A quotation of the original text passage containing the trait information for the current taxon |





| measurementDeterminedBy | Person who entered the trait information for this taxon into the database. |
| measurementDeterminedDate | Date the trait information was entered into the database or last modified. |

| | A | B | C | D | E | F | G | H | I | J | K | L | M | N | O | P | Q | R | S | T | U | V | W | X | Y | Z | AA | AB | AC | AD | AE | AF | AG |
|---|---|---|---|---|---|---|---|---|---|---|---|---|---|---|---|---|---|---|---|---|---|---|---|---|---|---|---|---|---|---|---|---|---|
| 1 | Taxon | Valid_name | S1 | S2 | S3 | S4 | S5 | W1 | W2 | W3 | W4 | W5 | BF1 | BF2 | BF3 | BF4 | BF5 | SK1 | SK2 | SK3 | SK4 | SK5 | F1 | F2 | F3 | SO1 | SO2 | SO3 | R1 | R2 | R3 | R4 | LD1 |
| 4 | Acanthonotozoma | Acanthonotozoma | | | | | | | | | | | | | 3 | | | | 3 | | | | | | | | | | | | | 3 | |
| 5 | Acanthonotozomatidae | Acanthonotozomatidae | | | | | | | | | | | | | 3 | | | | 3 | | | | | | | | | | | | | 3 | |
| 6 | Acanthostepheia | Acanthostepheia | | | | | | | | | | | | | 3 | | | | 3 | | | | | | | | | | | | | 3 | |
| 7 | Acanthostepheia malmgreni | Acanthostepheia malmgreni | | | | | | | | | | | | | 3 | | | | 3 | | | | | | | | | | | | | 3 | |
| 8 | Aceroides | Aceroides | | | | | | | | | | | | | 3 | | | | 3 | | | | | | | | | | | | | 3 | |
| 9 | Aceroides (Aceroides) | Aceroides (Aceroides) | | | | | | | | | | | | | 3 | | | | 3 | | | | | | | | | | | | | 3 | |
| 10 | Aceroides (Aceroides) latipes | Aceroides (Aceroides) latipes | 3 | | | | | | | | | | | | 3 | | | | 3 | | | 3 | | 3 | | | | | | | | 3 | |
| 11 | Acmaeidae | Acmaeidae | | | | | | | | | | | | | | 3 | | | | | | | | | | | | | | | | | | |
| 12 | Actiniaria | Actiniaria | | | | | | | | | | | | | | | | | | | 1 | 2 | | | 3 | | | | | | | | | |
| 13 | Adapedonta | Adapedonta | | | | | | | | | | | | | | 3 | | | | | | | | | | | | | | 3 | | | | 3 |
| 14 | Admete | Admete | | | | | | | | | | | | | | 3 | | | | | | | | | | | | | | | | | | |
| 15 | Admete viridula | Admete viridula | | | | | | | | | | | | | | 3 | | | | | | | | | | | | | | | | | | |
| 16 | Admetinae | Admetinae | | | | | | | | | | | | | | 3 | | | | | | | | | | | | | | | | | | |
| 17 | Aglaophamus | Aglaophamus | | | | | | | | | | | 3 | | | | | | | | | | 3 | | | | | | | 2 | | | 3 |
| 18 | Aglaophamus malmgreni | Aglaophamus malmgreni | | | | 3 | | | | | | | 3 | | | | | | | | | | 3 | 3 | | | | | | 3 | | | 3 |
| 19 | Akanthophoreidae | Akanthophoreidae | | | | | | | | | | | | | | | | 3 | | | | | | | | | | | | | | 3 | 2 |
| 20 | Akanthophoreus | Akanthophoreus | | | | | | | | | | | | | | | | 3 | | | | | | | | | | | | | | 3 | 2 |
| 21 | Akanthophoreus gracilis | Akanthophoreus gracilis | | | | | | | | | | | | | | | | 3 | | | | | | | | | | | | | | 3 | 2 |
| 22 | Alcyonidiidae | Alcyonidiidae | | | | | | | | | | | | | | | | | | | | | | | | | 3 | | | | | | |
| 23 | Alcyonidiina | Alcyonidiina | | | | | | | | | | | | | | | | | | | | | | | | | 3 | | | | | | |
| 24 | Alcyonidioidea | Alcyonidioidea | | | | | | | | | | | | | | | | | | | | | | | | | 3 | | | | | | |
| 25 | Alcyonidium | Alcyonidium | | | | | | | | | | | | | | | | | | | | | | | | | 3 | | | | | | |
| 26 | Alcyonidium gelatinosum | Alcyonidium gelatinosum | 3 | | | | | | | | | 3 | | | | | | | | | | | | | | | 3 | 1 | | | 2 | | | |
| 27 | Allantactis | Allantactis | | | | | | | | | | | | | | | | | | | | | | | | | | | | | | | | |

**Figure 5.** A clipping from the fuzzy coded trait matrix returned by the Arctic Traits Database when the "Data in matrix format" is chosen as export option from the download section. Species are rows ("Valid_name" refers to the currently accepted taxonomy in WoRMS), abbreviated trait categories are columns. For abbreviations of trait categories see Table 3. Due to the database structure zero codes ("0") are not displayed (see Table 6).

### 3.3 Database specification

The website runs on an Apache 2.2. server, the database is implemented in MySQL 5. PHP 5 is used as a scripting language. Web technologies used are HTML4, CSS and JavaScript/Jquery.

## 4 Results

### 4.1 Taxonomic data coverage

At present, the database contains 1211 Arctic marine benthic invertebrate taxa. Thereof 375 are on species level, 334 on genus level, and 212 on family level. The remaining 290 taxa are on higher taxonomic levels. The largest taxonomic group in the database at present stage are the Arthropoda with 357 taxa (88 entries on species level), followed by the Mollusca with 321 taxa (102 entries on species level) and the Annelida with 320 taxa (119 entries on species level) (Fig. 6).





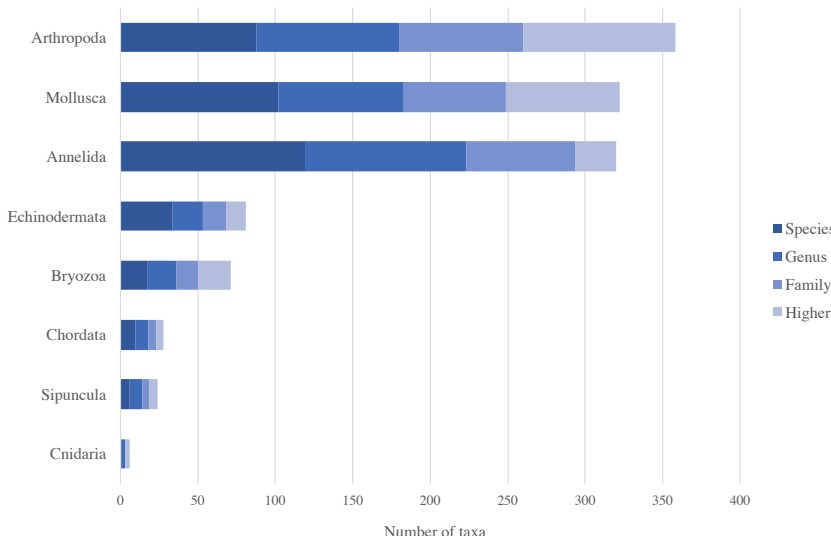

**Figure 6.** Taxonomic data resolution.

**4.2 Trait data coverage**

At present, the database contains 20 traits and 85 trait categories with in total currently 8107 entries of trait information. The trait for which most entries exist is "Skeleton" (1067 entries), followed by "Larval development"

(696 entries) and "Reproduction" (680 entries) (Fig. 7). The phylum with most entries are the Annelida (3614 entries, 45 %), followed by Arthropoda (1750 entries, 22 %) and Mollusca (1316 entries, 16 %). Regarding the taxonomic level, most trait information was added on the species level (45 %), less on the genus (23 %) and family level (18 %). The trait with fewest entries was Body weight (0.09 %), which will probably be removed from the database.




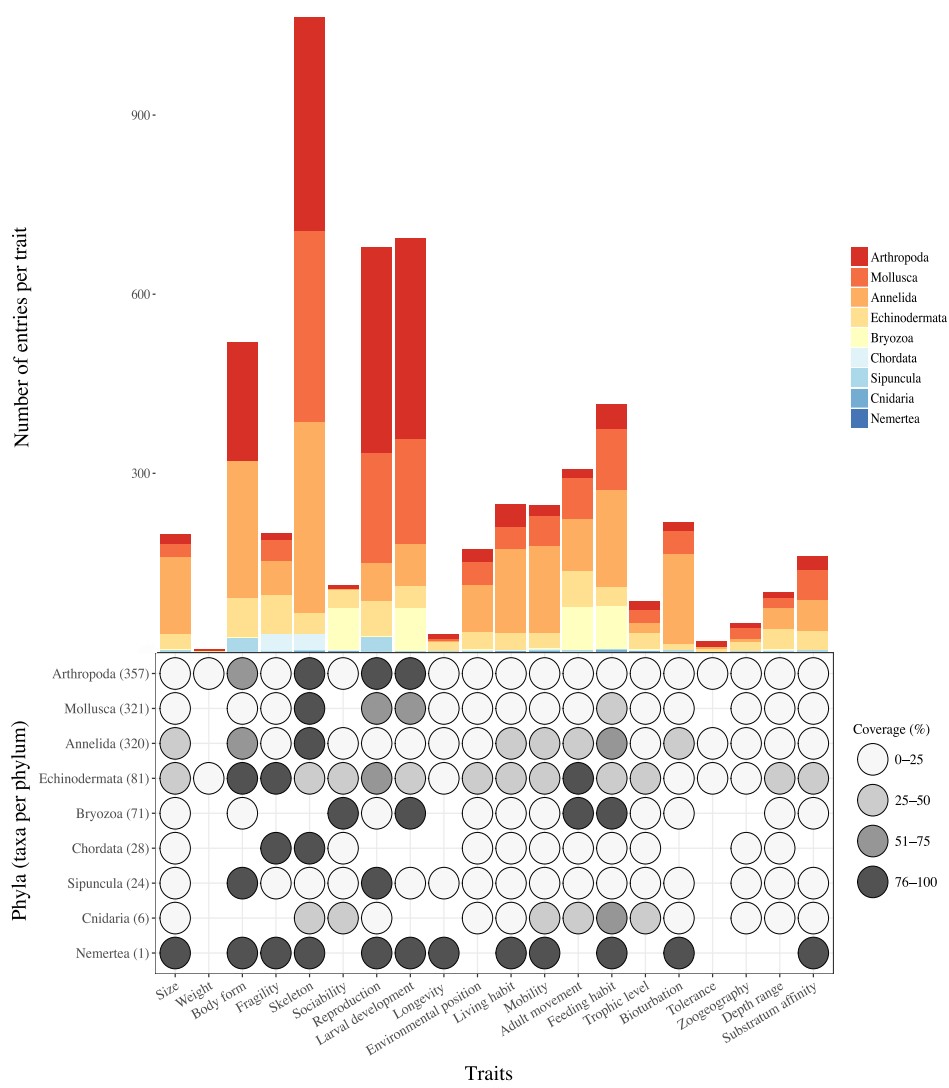

**Figure 7.** Scheme visualizing the data entries per trait (bar chart), the number of taxa per phylum (brackets), and the data coverage per trait per phylum (dot plot).

### 4.3 Bibliography

The Arctic Traits Database currently includes 198 sources of trait information. Thereof 57 % scientific papers, 15

295 % webpages, 12 % are books, and 6 % are expert communications and personal observation ("Other"). Most

separate sources were found for the phylum Echinodermata, followed by Arthropoda, and Annelida.



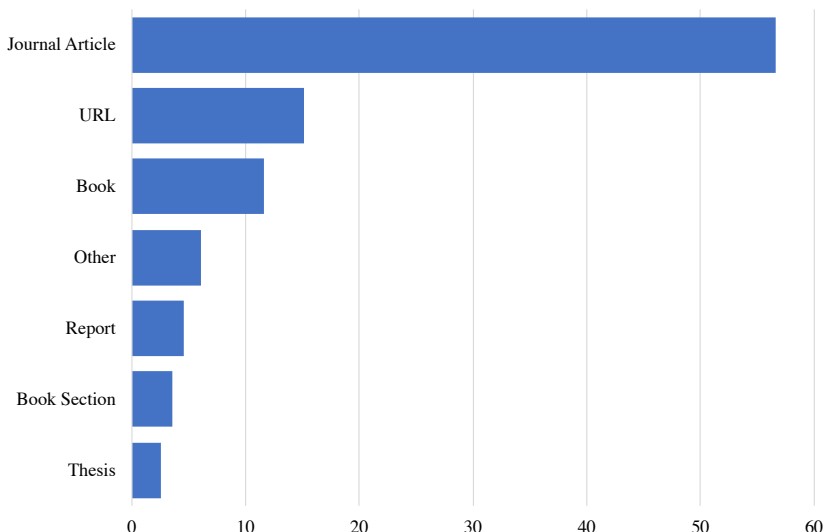

**Figure 8.** Relative amount (%) of trait source types.

**5 Discussion**

Although the Arctic Traits Database is still growing as new taxa and trait information are added, certain trends in data completeness or scarceness, respectively, became apparent (Fig. 7). Thus, the database is not only a valuable tool for collecting and providing information, but also for pointing out where more research might be needed. Regarding the 20 traits included at the present stage, it shows that our knowledge on e.g. the longevity of many Arctic benthic species is still limited (information only for < 1 % of species). This lack of data on species longevity

is astonishing, as polar taxa are traditionally depicted as slow growing and long-lived compared to their relatives from lower latitudes. Accordingly, one might have expected that more studies and measurements are available for a variety of Arctic taxa, which is not the case for many groups. Other traits that are currently underrepresented are maximum body weight and tolerance (both also <1 %).

        Regarding our interest to identify knowledge gaps, a special strength of the database is the implemented

flagging system (described in detail in the supplement). As registered users continue to upload trait information, also more "conflicts" – i.e. cases where the sources or observations added by different users point towards different trait categories – may arise. Such cases are then indicated by a red flag and can be easily filtered for. Monitoring and statistical evaluation of these cases will grant important information on where conflicts exist and for which taxa or traits future research is needed. Such evaluation will also aid to identify which traits are more robust (i.e.

are never flagged), and which show a higher plasticity (frequent flagging). This kind of information is of tremendous value as it can aid the choice as of which traits to include in prospective trait-based studies. Apart from clearly diverging source information, also different levels of experience or customs in fuzzy coding might lead to red flags in the system. Here the editorial team will take care for consistency by solving the conflicts according to the database standard, by that also fostering a standardized way of coding within the community. In addition,

repetitively occurring discrepancies in the coding of certain traits might also point towards a need for revision of





these trait categories or their definitions, or maybe even the adding of a new trait, in that way improving the quality of the database.

In addition to the above discussed knowledge gaps surrounding certain traits, also the data coverage among taxonomic groups varies considerable (Fig. 7). This potentially mirrors the sampling design of the underlying datasets. Some taxonomic groups such as the polychaetes clearly dominate many benthic soft-bottom communities, while other taxa such as the shrimp/caridea are highly mobile and might be permanently undersampled with sampling gears like grabs, box corers, or bottom trawls (Eleftheriou and McIntyre, 2007). This points toward the need to include also datasets derived from video and still image analysis in the future development of the database. These methods – despite certain disadvantages (discussed in Degen et al. 2018, Supplementary file 3) – have the great benefit that also traits of hard bottom communities can be analyzed, ecosystems which are at present stage underrepresented in the Arctic Traits Database.

## 6 Data availability

The Arctic Traits Database is hosted at the University of Vienna (Austria) and can be accessed via https://www.univie.ac.at/arctictraits/ (https://doi.org/10.25365/phaidra.49).

## 7 Conclusions

The Arctic Traits Database provides an easy accessible and sound knowledge base of traits of Arctic benthic invertebrates and will thus facilitate prospective trait-based studies for a variety of benthic ecologists at all career stages. Its sophisticated structure accounts for the most commonly raised demands to contemporary trait databases: 1) obligate traceability of information (every entry is linked to at least one source), 2) exchangeability among platforms (use of most common download formats), 3) standardization (use of most common terminology and coding scheme), and last but not least 4) user friendliness (granted by an intuitive web-interface and rapid and easy download options). The combination of these aspects makes the Arctic Traits Database a cutting-edge tool for (not only) the marine realm and a role-model for prospective databases.

## Author contribution

RD designed the project and performed the trait data collection. SF performed database and webpage development and design. RD prepared the manuscript with contributions from SF.

## Acknowledgements

The authors wish to thank all collaborators that support the Artic Traits Project, especially Bodil Bluhm, Jackie Grebmeier, Lauren Sutton and Dieter Piepenburg. This work was supported by the Austrian Science Fund (FWF; T 801-B29) to RD.

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
