# Peer review of "The Arctic Traits Database – A repository of arctic benthic invertebrate traits"

_Earth System Science Data, 2018_

## Referee Comment (RC1) · Anonymous Referee #1 · 20 Nov 2018

Review ESSD-2018-97, Arctic benthic traits

Interesting data approach, possibly a useful topic, seems like a good candidate for ESSD. Presentation however leaves much to be desired.

Thinking ahead (starting now) to urgent marine issues in the Arctic (loss of snow and ice, change from solid to liquid precip, changing run-off, changing local mixing and large scale circulation, change from predation to grazing, change in primary production / carbon fluxes / nutrient recycling, invasive species, increase in IUU fishing), and of the key role of benthic ecosystems in all the above, I think the authors intend to take an approach that says "document what we have from a functional approach so that we can better anticipate, monitor, detect and model on-going and future changes." Further "here we present a tool that can help our community achieve the functional approach". So far, so good, but how will this tool get used within but particularly beyond the benthos community. What changes, improvements, increased compliance, etc. does this tool need to serve valid research functions for the future Arctic? The authors hint at these directions and questions but give us only bibliometrics and screenshots?

About traits, authors emphasise, particularly in the introduction, usefulness of the trait approach as "indicators of ecosystem functioning" (authors words, page 1 line 30) for which they then elaborate: biodiversity, vulnerability to changing climate, etc. They assume that readers will accept the trait approach as somehow advantageous ("inherent advantage", page 1 line 33). Perhaps, but a reader reaches the end of this paper having encountered almost no examples (on page 11 we get one useful example of how fuzzy coding works for 'motile' species) of how the database, if fully populated, will help us address crucial issues. The authors seem to want to demonstrate a "popular" option while readers want and need to understand how this tool helps us address urgent research questions? Not how popular, how useful!

The Degen et al. 2018 paper in Ecological Indicators (open access, thanks) presents substantial sections on challenges and a specific roadmap. Without repeating verbatim, a précis of that message should find a home here, to set the stage? The authors repeatedly allude to this work meeting community needs and community standards. We could better accept those assurances if we had some tangible examples. Suggest a re-write along the lines of the following outline:

Introduce the trait approach to the earth sciences data community
- briefly justify trait approach compared to taxonomic approach,
- what one can do differently / better in terms of monitoring, ecosystem modelling, carbon or nutrient fluxes, etc.
- what more crucial place than coastal shelves of the Arctic.

Your Arctic Traits database
- goals
- approach
- content
- accessibility
- interoperability

Utility, both as an ingest tool and as a research tool

Contents so far

Example (1 or 2) how to use it
- something about biogeography, invasive or migration
- something about carbon and nutrient fluxes, number and clearance rate of filter feeders, how a benthic ecosystem in the Chukchi might respond to changed carbon imports with changed nitrogen returns, dependence of community structure, feeding activity, reproductive timing, nutrient fluxes on temperature and oxygen, differences Chukchi to Barents, etc. Real example or, if present data prove too limiting, hypothetical example.

What next?
- as an ingest tool and community repository

- as a research tool for a changing Arctic

In the view of this reviewer, the authors have sufficient information to provide, after revision, a much improved description of and guide to this database. Don't show us what we can find ourselves on the web page, show us how we can use this tool!

Brief specific comments, assuming the authors make a major revision as recommended:

The review apparently treats benthos as independent of water column, but what about sea ice cover, plankton particulate carbon deposition, carbon fluxes, historical depletion of whale and seal populations, continuing harvest of krill, etc. Give us please the valid benthos fully interactive with and essential to water column processes.

Page 2 line 36: Figure 1. Figure 1 not useful nor relevant. Because this reviewer mistrusts any topic where the authors must 'prove' its relevance by starting from bibliometric records, I suggest you simply leave it out.

What about Russian source materials. Kedra et al, cited, addresses this issue slightly and these authors reference Laptev Sea Lena R outflow transects work published by Kokarev et al. but, as for plankton, any database of Arctic ecology that does not include overt mechanisms to include Russian language publications will miss a very major fraction of possibly useful information? Does the benthos suffer a similar language barrier? If so, how will the authors address such barriers?

Page 2,3, Table 1: Good list but gets messy and out of order by the bottom entries. Include row demarcation? No diatom or coccolithophore (live or as deposited) databases? As for Figure 1, how useful is this table in a description of the particular Arctic benthic database? Leaving it out would not impact the overall description?

Page 3 line 62: "atomised"? A database term? Most readers will not know at this point what you mean by that word. This reviewer knows DarwinCore metadata guidance, but other readers will want a reference?

Page 3 lines 68 to 70: Agree, and this represents the strong motivation and potential impact of this work. Move this statement earlier, in a more prominent position?

Reference to a "pan-Arctic" approach and simultaneously, apparent regional focus (Svalbard, Chukchi)? In fact, we get no biogeographic information whatsoever from this database. Why this regional mention here that never gets a follow up?

Page 4 line 93: Costello et al represents a weak and not particularly reliable reference, mostly a self-citation tool for Costello. Fundamentally, Costello et al. recommend following the BIOTIC and FishBase database models. Do the authors not have something stronger on which to base their selections? One of the other marine species databases listed in Table 1, for example? Or other work that satisfies Steps 1 and 2 of the workshop report?

Page 4 line 95 "deep linked"? A database term? Reader does not know what the authors mean here?

Page 4 line 97: In GBIF a user can find reported occurrences of species by geographic location. As presented today, the Arctic Traits databases offers zero geographic location information. Reader will need to copy the species name from Artic Traits into GBIF to find location. I tried that for Nereis Linnaeus, more than 7000 records in GBIF including hundreds in the Arctic, but no zoogreographic information in Arctic Traits? Is this an example, not very successful, of "deep linked"? Should Arctic Traits become traits database linked under GBIF, for all co-listed taxa?

Page 4 line 103 to 108: confusing section! Physiological traits not defined nor well justified. Are they interesting or not interesting, retrievable or not retrievable. Are Arctic species generally eurythermal (which also depends on life history stage ) or stenothermal? Reader has no idea what to conclude from this section or about the inclusion or not of physiological traits in the database.

Page 10 line 155, 156:  Here readers learn that Arctic Traits database includes species with wider biogeographic ranges, not only those species with exclusively Arctic ranges.  This inclusion seems to relate to an earlier question of whether the function descriptions in the trait tables referred to only polar or to cosmopolitan species.  Apparently the latter?  Needs clarification!

Page 11, 12 fuzzy coding:  A necessary inclusion, well described, good use of examples!

I don't know ESSD policy, but most journals do not publish web page screenshots.  Give us links instead?  Here the authors unfortunately take the approach of showing us the product rather than demonstrating its utility.  Walk us through a couple examples, using links in place of screenshots?

Tooltip function (dragging cursor across indicator bar) does not work on my machine (MacBook Pro, OS 10.14 Mohave, Safari 12.0).

To get data I need to submit a request.  That means that Arctic Traits knows my IP address and can find my user information?

Downloaded skeleton file, largest category so far, very detailed, successful download, data access seems good.  But, now that I have it, how would I use it?  Find all the calcareous species to estimate their role in benthic carbon cycle?  I find almost 900 records, out of  2040 total, impressive.  After this initial sort I would need to resolve too-numerous species overlaps / redundancies?  E.g. 900 records might really only represent 500 or 600 valid independent species.  The database won't do this taxonomic clarification step automatically?  I assume in the database as opposed to the .csv file, I can click through to the exact reference and any text excerpts if I desire?  Next, on the carbon question, I would want to know sea floor population density of these calcareous organisms, carbon fixation rates as a function of season, temperature, O2, POC or DOC fluxes, biogeographic distribution including proximity to, for example, riverine inputs or ice fronts or ocean circulation fronts.  I might find helpful information under Body Weight, Living Habit, Reproduction, Feeding Habit, Tolerance, and Depth Range.  Zoogeographic here would provide zero useful information.  But, in general, I would or would not find useful information here?  As an alternative, for a species whose carbon uptake rates I knew from literature, I could go to GBIF to learn its frequency of occurrence in Arctic regions of interest and then do some spatial and physiologic extrapolations?  How did the Arctic traits database help me or hinder me in this case?  A weak example chosen on my part?  If so, give us a stronger more-favourable example?

---

## Referee Comment (RC2) · Anonymous Referee #2 · 20 Jan 2019

Review essd-2018-97 of The Arctic Traits Database – A repository of arctic benthic invertebrate traits, by Renate Degen & Sarah Faulwetter

This is a comprehensive and impressive Trait compilation that deserves honor and gratitude and a great "thank you" for leading this and compiling this together with other scientists.

This is, as indicated, a great start because .. "Traits can be analyzed across wide geographical ranges and across species pools (Bernhardt-Römermann et al., 2011), they can be used to calculate a variety of functional diversity indices (Schleuter et al., 2010), to estimate functional redundancy, or be used as 30 indicators of ecosystem

functioning (Bremner et al., 2006). Given the rapid changes we observe in many marine regions of the world, and especially in the Arctic Ocean (Wassmann et al., 2011), the potential to indicate vulnerability to climate change and biodiversity loss, or to estimate climate change effects on ecosystem functions is another inherent advantage of trait-based approaches.

I find the potential of the trait data as being very useful, but this database and the methods and materials is under development and might also need more detailed descriptions.

1. Check the data quality: I find the dataset a good establishment for an ongoing and continues work. Error estimates and sources of errors need to be more clearly expressed when traits-values/categories are missing for, particularly, many Arctic species. Processing of the traiting in the further analyses and presentations need to be elaborated (see notes). 2. Consider article and data set: I find the traiting of species of high quality and this important work is based on much effort. 3. Check the presentation quality: The species information is highly useable for the traits/modules given in the Arctic Trait Database and of high quality. But if a given dataset has species without defined traits, it is more uncertain what to do.

Rating Reviewers are asked to decide how well the respective data sets presented by an article and the article itself meet the following criteria = 2 - 3 Significance  Uniqueness: The Arctic Traits Database is a unique. It has and will continue to compile traits data that shall be used on a general basis, and allow comparisons areas regions. This has been a huge work and therefore not possible to replicate on a routine basis.  Usefulness: The traits in the Arctic Traits Database might, or might not be, used in future works depending on how many species are found both in the work that will be implementing the trait data and in the Trait database itself.  Completeness: The Arctic Trait database is a developing product that will evolve in the same paste as the development of biological data of particularly Arctic species.  Data quality The available species/taxon trait-data are readily presented and accessible for use. What

type of analyses and how to deal with missing trait data are not given, and need to be developed by the user.

Presentation quality Categorical traits and continuous traits: The categorical traits (e.g. body shape, reproduction, larval development, and more) are well suited to be divided into modalities and to be used in fussy coding. Here obligate traceability of literature information is important, but lacking for, as said in the manuscript, many Arctic species and rocky bottom communities. Please explain how to work with trait based analyses when some (many) Arctic species/taxons cannot be traitet due to lack of information. In other words: how large part of a database can lack information, but still be possible to analyze? Is it 5%, 15%, 50%? Or is it so, that the use of the Arctic Trait Database (Degen & Faulwetter) cannot fulfill its purpose before all categorical trait information is in place and with the obligate traceability of literature information? Please explain how the user can work around the problem with categorical trait information that lacks the obligate traceability of literature information.

The continuous traits (e.g. body weight, size, height) are measures that are (or could be) obtained during the field work, and are therefore not be limited to "the obligate traceability of literature information". You mention in the manuscript that "Arctic species" can be deferent than their relatives from lower latitudes". With this, you open for a discussion on species "plasticity" and "adaptation" from area to area. Please explain the difference between a "Trait value" and a "field value" here. If "field values" are to be lumped into broad categories (see table 3: e.g. body size, body weight, zoogeography – i.e. tolerance of temperatures, and depth range), they might be a tool to compare across areas (if same sampling tool has been used). But if used as a long term monitoring assessment, the "trend" will be "drowning" inside the category, and most likely a catastrophe needs to happen before a signal come forward. With other words, the "early warning signal" will not be available unless the field data are used as detailed as possible. I ask the auditors to mention this in the manuscript in order to make it more clear what the purpose of the traits is. In the arena of "plasticity" and

"adaptation" from area to area, there might be many different values given by a variety of literature references. You have used a "flagging system" to observe this type of "inconsistency". Would it be an idea to simply accept that some values are not consistent and therefore need to be obtained as field values, and not be added to the Arctic Trait Database? If not, please explain what to do with such continuously variable with many different literature-based values – do you trust your field data, or do you need to take the "global plasticity" into consideration? If this depends on your scientific question, please explain this carefully, so the reader will understand the differences between and the usefulness of a "trait-value" and a "field-value". Another point that you might bring forward is "species more or less affected by trawling (see Body size in tab 3)". What type of Body size value are you referring to here: • the station mean body size of the species • the area mean body size of a species • or "the obligate traceability of literature information value" These three values might differ a lot. If your answer is that "this depends on your scientific question" – then please explain this carefully to the reader. Please also be aware that a trawl-vulnerable organism (e.g. a sea feather or similar) might evaluated by its "body-size". But, is a small (i.e. young) individual of a species "less vulnerable toward effects from trawling" than a full-grown individual of the same species? Is it so that in this type of vulnerability assessment studies a "obligate traceable literature information value" is more correct? Because it is not clear forward when to use field data or when to use "obligate traceable literature information value" it is very important that a pre-evaluation period is made before trait analyses are made.

The use of the data and the visual outcome As written in the Introduction: Traits can be analyzed across wide geographical ranges and across species pools (Bernhardt-Römermann et al., 2011), they can be used to calculate a variety of functional diversity indices (Schleuter et al., 2010), to estimate functional redundancy, or be used as indicators of ecosystem functioning (Bremner et al., 2006). Given the rapid changes we observe in many marine regions of the world, and especially in the Arctic Ocean (Wassmann et al., 2011), the potential to indicate vulnerability to climate change and biodiversity loss, or to estimate climate change effects on ecosystem functions is another

inherent advantage of trait-based approaches. There is an issue of "not all species (particularly Arctic species) being traited" due to the lack of "obligate traceable literature information" on morphological, life history, behavioral traits information. I therefore wonder what type of method need to be used to calculate and finally obtain a result that can be, for example, depicted on a map. Again, I expect that the answer will be that this "depends on the scientific question that is been asked". But could you please explain how to move from an incomplete species-traits database to the most appropriate method, and further to the presentation (be it a map or a figure) that identify the results but also the flaws?

General comments: 1) Line 39: what is a species "trait" – please define very clearly. 2) As clearly stated by the auditors – few (if any) literature proved traits are available for all species – what do you do with a field analyses when you have "missing trait data" because of lacking literature evidence? 3) Even if one or more literature references are available for a species Trait – how can we be sure that its correct? –the wording in line 13 "- obligate traceability of information (every entry is linked to at least one source)" seems overemphasized (see also line 157-158). When a literature based trait is not possible to find for a species, and when a specific trait (for example "size") can vary from one geographic area to another, a "obligate literature traceability" can be misleading information. 4) Line 86-88: traits used in previous studies and databases should be favored to enable comparisons across studies (Degen et al. 2018), and 3) the traits should be usable across a wide geographical area (Bremner et al. 2006). Characteristics such as "body weight", "size", "morphology", "temperature preference", and "depth range", are area depended and subjected to adaptivity/plasticity for existing environment. Do to the "line 13 - obligate traceability of information (every entry is linked to at least one source)" meaning that your species need to be pre-defined with a trait, please explain why changes in "field-based traits" such as "size" cannot be used. 1) Field based characteristics are very important "traits" in monitoring to detect "early warning signals" (see also line 70). These type of field based traits cannot be useful if lumped into large modalities as for example S1-S5 and W1-W5 because a

"catastrophe" need to happen before a species will fall from one modality to another. And if literature based evidence shall be given for a trait, such as "size" – how can we detect a change in an area? Please explain why "field-based traits" are not coming into consideration in your choice of traits (line 85-88)? 2) Line 85-88 – says that chosen traits shall be available and applicable to all benthic taxa . . . but still . . . as also stated in the manuscript – not all species (particularly not the Arctic species) can be traitet due to the lack of literature based evidence – so what do you do when you will like to apply traits to a dataset? Does this means that the trait database cannot be used because of this lack of literature based trait evidence? Please explain how to cope with this. 3) Traits must be selected in accordance to the scientific question. If a comparison to other areas are important not only same trait and modality has to be used, but also the same type of analyses. Please describe the steps from the fuzzy coding, the analyses and the mapping that can be applied to "all" in order to compare across regions.

Line 115-148: You divide your traits into "indicators of ecosystem functions (effect traits)" and "changes in the environment (response traits)". But it is unclear how you use these two categories in table 3. Can you please mention them specifically here? Please make it clear in Table 3 if Body Size, Body Weight, Zoogeography, Depth Range are "Field-based Traits" or if they have to be "obligate traceability of information (every entry is linked to at least one source)". Table 3: Please define the "Zoogeography" Trait. What is an Arctic, a Arctic-boreal, and a boreal species defined as. . . what temperature ranges? If lines 149-154 is true, please add them up front in the paper so it is clear that a "trait" have to be both "field-based" and "literature-based" in order to be able to apply a traitbased analyses on all, and not only on a subset of species. Chap 4.2 – it is also obvious that the trait database most efficiently covers macrofaunal species as Annelida, Arthropoda, Mollusca .. It might be written up front in the manuscript that mega-fauna, such as Echinodermata, Sponges are "under development" and therefore not fully operative due to many missing literature based-traits. Line 188: I agree that "Body weight" can be removed from the database because this trait is plastic and variable within and between areas, i.e. not a global constant, and will not be covered

efficiently by 1 or 2 literature based references.

---

## Author Comment (AC1) · 1 Feb 2019

**Author's response**

We would like to thank the two anonymous reviewers, whose comments significantly improved our manuscript. Please find our detailed response below. We added numbering to the reviewers comments to easier link to responses that apply several times. Corresponding changes in the manuscript are highlighted in yellow. Line numbers refer to the revised manuscript, attached to this response letter.

Larger changes in manuscript:

- Trait body weight removed from database (see comment 24, reviewer 2).
- All figures, tables and numbers in the text are updated to the latest status of the database.
- Comments of reviewer 1
   → Response to reviewer 1 in blue
- 2) Comments of reviewer 2
  → Response to reviewer 2 in green.
- 3) Revised manuscript
- 4) Revised supplement

**Reviewer 1**

Interesting data approach, possibly a useful topic, seems like a good candidate for ESSD. Presentation however leaves much to be desired.

1) Thinking ahead (starting now) to urgent marine issues in the Arctic (loss of snow and ice, change from solid to liquid precip, changing run-off, changing local mixing and large scale circulation, change from predation to grazing, change in primary production / carbon fluxes / nutrient recycling, invasive species, increase in IUU fishing), and of the key role of benthic ecosystems in all the above, I think the authors intend to take an approach that says "document what we have from a functional approach so that we can better anticipate, monitor, detect and model on-going and future changes." Further "here we present a tool that can help our community achieve the functional approach". So far, so good, but how will this tool get used within but particularly beyond the benthos community. What changes, improvements, increased compliance, etc. does this tool need to serve valid research functions for the future Arctic? The authors hint at these directions and questions but give us only bibliometrics and screenshots?

About traits, authors emphasise, particularly in the introduction, usefulness of the trait approach as "indicators of ecosystem functioning" (authors words, page 1 line 30) for which they then elaborate: biodiversity, vulnerability to changing climate, etc. They assume that readers will accept the trait approach as somehow advantageous ("inherent advantage", page 1 line 33). Perhaps, but a reader reaches the end of this paper having encountered almost no examples (on page 11 we get one useful example of how fuzzy coding works for 'motile' species) of how the database, if fully populated, will help us address crucial issues. The authors seem to want to demonstrate a "popular" option while readers want and need to understand how this tool helps us address urgent research questions? Not how popular, how useful! We understand the reviewers interest in specific applications, however, this is not in the scope of the present paper, nor is it – to our understanding – within the aims and scope of ESSD ("Any interpretation of data is outside the scope or regular articles"). RD has a publication that shows some specific applications (building on the data in the database) in preparation (more information can be found on the webpage of the Arctic Traits Project https://sites.google.com/site/arctictraits/home/the-project). But the present manuscript is a data paper. It describes the Arctic Traits Database: how it was developed, the improvements compared to other data repositories, how we tackle standardization issues, what it contains, how it can be used (i.e. how the trait data can be accessed and downloaded). How users proceed from there on is beyond the scope of this work. Trait-based approaches are not new, they are used in marine ecology already since the late 1970ies (as we show in Fig. 1). Consequently a plethora of research questions that can be tackled and applications exists. The only thing they all have in common is that the basic input are traits (see manuscript line 40). Although we cannot go into methodical detail here, we do refer to some concrete examples right at the begin of the introduction (line 28 f) which guide the interested (but yet unaware) reader further. We now added some more references here (Darr et al. 2014, Foden et al. 2013, Hewitt et al. 2016), to further stress the methodical variety. We now also added references of papers that focus specifically on methods (Beauchard et al. 2017, Kleyer et al.2012 ) in line 40.

Although the main users of our database are (and will be) mainly benthic ecologists, the data is of use also to other disciplines. These include climate researchers, ecosystem modelers, oceanographers, biogeographers, and potentially even geologists.

Further the code package underlying this database is now accessible at figshare via https://doi.org/10.6084/m9.figshare.7491869. This allows scientists to easily build their own trait database for other ecosystem components (e.g. zooplankton, phytoplankton, marine mammals, ...). We added this information in part 3.3, line 272f.

2) The Degen et al. 2018 paper in Ecological Indicators (open access, thanks) presents substantial sections on challenges and a specific roadmap. Without repeating verbatim, a précis of that message should find a home here, to set the stage? The authors repeatedly allude to this work meeting community needs and community standards. We could better accept those assurances if we had some tangible examples. Suggest a re-write along the lines of the following outline:

Introduce the trait approach to the earth sciences data community

- briefly justify trait approach compared to taxonomic approach,

- what one can do differently / better in terms of monitoring, ecosystem modelling, carbon or nutrient fluxes, etc.

- what more crucial place than coastal shelves of the Arctic.

- Your Arctic Traits database
- goals
- approach
- content
- accessibility
- interoperability

Utility, both as an ingest tool and as a research tool Contents so far

Example (1 or 2) how to use it

- something about biogeography, invasive or migration

- something about carbon and nutrient fluxes, number and clearance rate of filter feeders, how a benthic ecosystem in the Chukchi might respond to changed carbon imports with changed nitrogen returns, dependence of community structure, feeding activity, reproductive timing, nutrient fluxes on temperature and oxygen, differences Chukchi to Barents, etc. Real example or, if present data prove too limiting, hypothetical example. What next?

- as an ingest tool and community repository

- as a research tool for a changing Arctic

In the view of this reviewer, the authors have sufficient information to provide, after revision, a much improved description of and guide to this database. Don't show us what we can find ourselves on the web page, show us how we can use this tool!

See response above (1) regarding the specific examples. Regarding the overall structure we follow an outline that is comparable to other database papers in ESSD (e.g. Brun et al. 2017):

- 1) Introduction & goals
- 2) Data (choice of taxa and traits)
- 3) Database (way of structuring and presenting data)
- 4) Results (current content of the database)

Brief specific comments, assuming the authors make a major revision as recommended:

3) The review apparently treats benthos as independent of water column, but what about sea ice cover, plankton particulate carbon deposition, carbon fluxes, historical depletion of whale and seal populations, continuing harvest of krill, etc. Give us please the valid benthos fully interactive with and essential to water column processes.

Our review paper (Degen et al. 2018 Ecological indicators 91: 722-736) was already reviewed and published in early 2018. At no point it states that benthos is independent from water column processes.

4) Page 2 line 36: Figure 1. Figure 1 not useful nor relevant. Because this reviewer mistrusts any topic where the authors must 'prove' its relevance by starting from bibliometric records, I suggest you simply leave it out.

We understand that figures are always a matter of taste. However, we consider Figure 1 relevant as it clearly shows the increased interest in biological traits, especially in studies from the benthic realm. As such it underlines the current need for sound trait databases that we stress in the introduction.

5) What about Russian source materials. Kedra et al, cited, addresses this issue slightly and these authors reference Laptev Sea Lena R outflow transects work published by Kokarev et al. but, as for plankton, any database of Arctic ecology that does not include overt mechanisms to include Russian language publications will miss a very major fraction of possibly useful information? Does the benthos suffer a similar language barrier? If so, how will the authors address such barriers?

This is definitely an important issue. We are happy to have now (since very recently) Valentin Kokarev on board the editorial team of the Arctic Traits Database, who will in future add information from publications in Russian language. 6) Page 2,3, Table 1: Good list but gets messy and out of order by the bottom entries. Include row demarcation? No diatom or coccolithophore (live or as deposited) databases? As for Figure 1, how useful is this table in a description of the particular Arctic benthic database? Leaving it out would not impact the overall description?

The purpose of this table is to give an overview of existing databases and the ways that data can be accessed, as such it helps to identify the improvements we offer with the Arctic Traits Database (i.e. online browsing + several download options like fuzzy coded trait matrix). We added the coastal phytoplankton trait collection by Riina Klais to the table. We also added row demarcations.

7) Page 3 line 62: "atomised"? A database term? Most readers will not know at this point what you mean by that word. This reviewer knows DarwinCore metadata guidance, but other readers will want a reference?

We changed the sentence in line 64 to "...and provide download of trait data in different tabular formats (i.e. data in columns, once following a database-specific format and once DarwinCore). " A reference for DarwinCore is added (Wieczorek et al. 2012).

8) Page 3 lines 68 to 70: Agree, and this represents the strong motivation and potential impact of this work. Move this statement earlier, in a more prominent position? Added now to the abstract in line 18 ("...including for the first time the option to download a fuzzy coded trait matrix").

9) Reference to a "pan-Arctic" approach and simultaneously, apparent regional focus (Svalbard, Chukchi)? In fact, we get no biogeographic information whatsoever from this database. Why this regional mention here that never gets a follow up? As stated in line 80: "The regional coverage currently comprises the Chukchi Sea and the Svalbard area". So this is so far as we got by now, but more data are added successively. This database provides species-specific information, not biogeographic information. For this we we refer to OBIS and GBIS (see line 102f).

10) Page 4 line 93: Costello et al represents a weak and not particularly reliable reference, mostly a self-citation tool for Costello. Fundamentally, Costello et al. recommend following the BIOTIC and FishBase database models. Do the authors not have something stronger on which to base their selections? One of the other marine species databases listed in Table 1, for example? Or other work that satisfies Steps 1 and 2 of the workshop report? We consider this paper important and the appropriate reference here, as it is the first that clearly states the importance of standardization processes and prioritizes the development of a marine trait database. The authors are all acknowledged experts in the field (along Marc Costello there is e.g. Leen Vandepitte from WoRMS and Harvey Tyler-Walters from BIOTIC).

**11) Page 4 line 95 "deep linked"? A database term? Reader does not know what the authors mean here?**

Deep linking refers to the use of a hyperlink that links to a specific web content. We changed to "...every species in the database is bidirectionally deep linked (i.e. connected via a hyperlink) to the World register of Marine Species..." in line 100.

12) Page 4 line 97: In GBIF a user can find reported occurrences of species by geographic location. As presented today, the Arctic Traits databases offers zero geographic location

information. Reader will need to copy the species name from Artic Traits into GBIF to find location. I tried that for Nereis Linnaeus, more than 7000 records in GBIF including hundreds in the Arctic, but no zoogreographic information in Arctic Traits? Is this an example, not very successful, of "deep linked"? Should Arctic Traits become traits database linked under GBIF, for all co-listed taxa?

See before (9). We are now linked with WoRMS (in both directions, since January 19 WoRMS provides deeplinks back to us). Traits data is outside the scope of OBIS and GBIF, as these two are biogeographic databases. They do not foresee the integration of trait data.

13) Page 4 line 103 to 108: confusing section! Physiological traits not defined nor well justified. Are they interesting or not interesting, retrievable or not retrievable. Are Arctic species generally eurythermal (which also depends on life history stage ) or stenothermal? Reader has no idea what to conclude from this section or about the inclusion or not of physiological traits in the database.

We don't really see what causes the confusion in this part, as we clearly state why physiological traits (examples thereof are given in brackets in line 111) are excluded. As stated, it relates to the violation of the preconditions for a trait to be included (i.e. being retrievable for most taxa and being usable across a wide geographical area). The preconditions are clearly explained just above (in line 88).

14) Page 10 line 155, 156: Here readers learn that Arctic Traits database includes species with wider biogeographic ranges, not only those species with exclusively Arctic ranges. This inclusion seems to relate to an earlier question of whether the function descriptions in the trait tables referred to only polar or to cosmopolitan species. Apparently the latter? Needs clarification!

We don't understand this question. In the revised manuscript (according to a comment by Reviewer 2) we added a definition to the trait categories of zoogeography in table 3 – does this solve the issue?

Page 11, 12 fuzzy coding: A necessary inclusion, well described, good use of examples!

15) I don't know ESSD policy, but most journals do not publish web page screenshots. Give us links instead? Here the authors unfortunately take the approach of showing us the product rather than demonstrating its utility. Walk us through a couple examples, using links in place of screenshots?

To our knowledge screenshots are tolerated in ESSD. Also, the screenshot in Figure 2b and those in the Supplement come from the restricted area of the database (access only for registered users), so this cannot be provided as web link.

16) Tooltip function (dragging cursor across indicator bar) does not work on my machine (MacBook Pro, OS 10.14 Mohave, Safari 12.0). Fixed that.

17) To get data I need to submit a request. That means that Arctic Traits knows my IP address and can find my user information?

Yes, the user submits a request and the IP is sent to the server. This is however already done when the user only views the site, not only when a request is submitted (normal HTTPS communication). The website is compliant with the European General Data Protection

Regulation (GDPR). While the IP address could theoretically be linked to identifying information, the University of Vienna does not exploit this information or use it in any other way (e.g. marketing) than ensuring the functionality of the servers and IT infrastructure. The University of Vienna, to be compliant with Austrian legislation (Art. 6 Abs. 1 lit. f DSGVO), may retain the information for maximally 30 days. This is already stated on the website at https://www.univie.ac.at/arctictraits/privacy.

Thus, the website, being hosted by an Austrian institution is compliant with Austrian laws and technical specifications are being implemented accordingly by the University of Vienna.

18) Downloaded skeleton file, largest category so far, very detailed, successful download, data access seems good. But, now that I have it, how would I use it? Find all the calcareous species to estimate their role in benthic carbon cycle? I find almost 900 records, out of 2040 total, impressive. After this initial sort I would need to resolve too-numerous species overlaps / redundancies? E.g. 900 records might really only represent 500 or 600 valid independent species. The database won't do this taxonomic clarification step automatically? We thank the reviewer for highlighting an important aspect, and now include also the rank of a taxon in the download (see also according changes in Table 7 and 8). Apart from that, the common scenario with trait databases is that users have a specific dataset (e.g. benthos abundance data of xy sample stations), for which they want to find trait information. So they upload their specific taxon list and download only the traits of exactly these taxa. Accordingly, for those users a taxonomic "clarification step" is not necessary.

19) I assume in the database as opposed to the .csv file, I can click through to the exact reference and any text excerpts if I desire?

The literature sources and excerpts are also included in the .csv download, so the user is not required to click manually through the database (although this option exists).

20) Next, on the carbon question, I would want to know sea floor population density of these calcareous organisms, carbon fixation rates as a function of season, temperature, O2, POC or DOC fluxes, biogeographic distribution including proximity to, for example, riverine inputs or ice fronts or ocean circulation fronts. I might find helpful information under Body Weight, Living Habit, Reproduction, Feeding Habit, Tolerance, and Depth Range. Zoogeographic here would provide zero useful information. But, in general, I would or would not find useful information here? As an alternative, for a species whose carbon uptake rates I knew from literature, I could go to GBIF to learn its frequency of occurrence in Arctic regions of interest and then do some spatial and physiologic extrapolations? How did the Arctic traits database help me or hinder me in this case? A weak example chosen on my part? If so, give us a stronger more favourable example?

As stated above (1), specific examples are outside the scope of this article type for this journal. But to follow up on the reviewers thought, one idea that comes to mind here are Brey's empirical models that can be used to estimate secondary production or respiration (Brey 2012). Required input are temperature, depth, body mass, and certain traits such as environmental position, mobility, feeding type (traits all included in the Arctic Traits Database). Other large-scale approaches (up to global) use trait information in combination with distribution data from OBIS or the IUCN red list. As one specific example, Foden et al. (2013) followed this approach to identify "The world's most climate vulnerable species" and used traits such as dispersal potential or habitat specialization.

- Brey T, 2012. A multi-parameter artificial neural network model to estimate macrobenthic invertebrate productivity and production. Limnology and Oceanography Methods 10: 581-589. DOI: 10.4319/lom.2012.10.581
- Brey T, 2001. Population dynamics in benthic invertebrates. A virtual handbook. http://www.thomas-brey.de/science/virtualhandbook/
- Foden, W. B., Butchart, S. H. M., Stuart, S. N., Vié, J. C., Akçakaya, H. R., Angulo, A., DeVantier, L. M., Gutsche, A., Turak, E., Cao, L., Donner, S. D., Katariya, V., Bernard, R., Holland, R. A., Hughes, A. F., O'Hanlon, S. E., Garnett, S. T., Şekercioğlu, Ç. H. and Mace, G. M.: Identifying the world's most climate change vulnerable species: a systematic trait-based assessment of all birds, amphibians and corals, edited by S. Lavergne, PLoS One, 8(6), e65427, doi:10.1371/journal.pone.0065427, 2013.

**Reviewer 2**

1) Please explain how to work with trait based analyses when some (many) Arctic species/taxons cannot be traitet due to lack of information. In other words: how large part of a database can lack information, but still be possible to analyze? Is it 5%, 15%, 50%? Most specific software (e.g. R package ade4) can deal with missing trait information, but it remains to the user who interprets the results to decide how many gaps can be tolerated, or whether a trait-based approach might no longer be reasonable. So far, there is no general acknowledged procedure on this. Some papers deal specifically with this issue, e.g. Tyler et al. (2012) or Májeková et al. (2016).

- Tyler, E. H. M., Somerfield, P. J., Berghe, E. Vanden, Bremner, J., Jackson, E., Langmead, O., Palomares, M. L. D. and Webb, T. J.: Extensive gaps and biases in our knowledge of a well-known fauna: Implications for integrating biological traits into macroecology, Glob. Ecol. Biogeogr., 21(9), 922–934, doi:10.1111/j.1466-8238.2011.00726.x, 2012.
- Májeková, M., Paal, T., Plowman, N. S., Bryndová, M., Kasari, L., Norberg, A., Weiss, M., Bishop, T. R., Luke, S. H., Sam, K., Le Bagousse-Pinguet, Y., Lepš, J., Götzenberger, L. and De Bello, F.: Evaluating functional diversity: Missing trait data and the importance of species abundance structure and data transformation, PLoS One, 11(2), 1–17, doi:10.1371/journal.pone.0149270, 2016.

2) The categorical traits (e.g. body shape, reproduction, larval development, and more) are well suited to be divided into modalities and to be used in fussy coding. Here obligate traceability of literature information is important, but lacking for, as said in the manuscript, many Arctic species and rocky bottom communities. Please explain how to work with trait based analyses when some (many) Arctic species/taxons cannot be traitet due to lack of information. In other words: how large part of a database can lack information, but still be possible to analyze? Is it 5%, 15%, 50%? Or is it so, that the use of the Arctic Trait Database (Degen & Faulwetter) cannot fulfill its purpose before all categorical trait information is in place and with the obligate traceability of literature information? See above (1).

3) Please explain how the user can work around the problem with categorical trait information that lacks the obligate traceability of literature information. The Arctic Trait Database does not contain trait information that is not backed up either by literature, database links, or in case of "personal observation" or "expert comment" by a contact that provided the information/observation. See part 2.3, line 152 f.

4) The continuous traits (e.g. body weight, size, height) are measures that are (or could be) obtained during the field work, and are therefore not be limited to "the obligate traceability of literature information". You mention in the manuscript that "Arctic species" can be

deferent than their relatives from lower latitudes". With this, you open for a discussion on species "plasticity" and "adaptation" from area to area. Please explain the difference between a "Trait value" and a "field value" here.

Due to our third precondition for traits to be included in our database ("the traits should be usable across a wide geographical area", part 2.2) we do not account for regional plasticity. Instead we use the maximum body size that we find in literature (or have measured in the field, or are informed of via communication with experts) (as previously suggested by e.g. Costello et al. 2015). To make this more clear to the readers this aspect is now explained more in detail in part 2.2 line 91f ("In order to fulfil this last precondition, the trait body size is provided as "maximum body size as adult" (see also Table 3). While clearly a tradeoff in regard to the detection of intraspecific plasticity, it enables the use of this trait across large spatial scales.").

5) If "field values" are to be lumped into broad categories (see table 3: e.g. body size, body weight, zoogeography – i.e. tolerance of temperatures, and depth range), they might be a tool to compare across areas (if same sampling tool has been used). But if used as a long term monitoring assessment, the "trend" will be "drowning" inside the category, and most likely a catastrophe needs to happen before a signal come forward. With other words, the "early warning signal" will not be available unless the field data are used as detailed as possible.

As the reviewer correctly states, the intraspecific plasticity (e.g. in body size) cannot be tackled with this database (see 4). However, users can still use our database to detect changes or "early warning signals" in their time series community data, as changes in the abundances and/or in the community composition will be reflected in the size spectra (not on species level, but on community level).

6) In the arena of "plasticity" and "adaptation" from area to area, there might be many different values given by a variety of literature references. You have used a "flagging system" to observe this type of "inconsistency". Would it be an idea to simply accept that some values are not consistent and therefore need to be obtained as field values, and not be added to the Arctic Trait Database? If not, please explain what to do with such continuously variable with many different literature-based values – do you trust your field data, or do you need to take the "global plasticity" into consideration? If this depends on your scientific question, please explain this carefully, so the reader will understand the differences between and the usefulness of a "trait-value" and a "field-value".

As stated above (4, 5), we deal with the obvious plasticity in body size by using the max. body size as adult (see Table 3). The "global plasticity" is an issue we briefly discuss in part 2.3 line 159f. Where possible we use literature from the Arctic, but when not available also wider sources are included. As for all traits, literature and/or field measurements are taken into account when available (as explained in part 2.3). For the final coding of the trait, the largest size value is considered (no matter if source is a book or field data). This holds true only on species level, in higher taxonomic levels (e.g. family) the broader range (if applying) is accounted for by coding a "2" over several size classes.

7) Another point that you might bring forward is "species more or less affected by trawling (see Body size in tab 3)". What type of Body size value are you referring to here: the station mean body size of the species, the area mean body size of a species or "the obligate traceability of literature information value" These three values might differ a lot.

**See above, 4-6.**

8) Please also be aware that a trawl-vulnerable organism (e.g. a sea feather or similar) might evaluated by its "body-size". But, is a small (i.e. young) individual of a species "less vulnerable toward effects from trawling" than a full-grown individual of the same species? We use max. body size as adult. Juvenile traits are not considered in this database except for larval development. A trade-off in favor to the goal of this database to be applicable across wider geographical areas. Users need to carefully assess which traits and categories that we provide are suitable for their analyses. As the literature reference is given, the user is free to trace back the information and re-evaluate the trait for their analyses, if needed.

9) Is it so that in this type of vulnerability assessment studies a "obligate traceable literature information value" is more correct? Because it is not clear forward when to use field data or when to use "obligate traceable literature information value" it is very important that a pre-evaluation period is made before trait analyses are made. See before, 4-6.

10) There is an issue of "not all species (particularly Arctic species) being traited" due to the lack of "obligate traceable literature information" on morphological, life history, behavioral traits information. I therefore wonder what type of method need to be used to calculate and finally obtain a result that can be, for example, depicted on a map. Again, I expect that the answer will be that this "depends on the scientific question that is been asked". But could you please explain how to move from an incomplete species-traits database to the most appropriate method, and further to the presentation (be it a map or a figure) that identify the results but also the flaws?

For a study that focuses only on a subset of species or traits, the data we provide may well be complete. And if not, it is up to the user to decide how to deal with the missing data, either by inference from related species, higher taxonomic levels, by using additional direct measurements, by conducting their own literature research to complement the information, or through statistical imputation methods.

**General comments:**

11) Line 39: what is a species "trait" – please define very clearly. Modified line 26 (first mention of the term) to make more clear.

12) As clearly stated by the auditors – few (if any) literature proved traits are available for all species – what do you do with a field analyses when you have "missing trait data" because of lacking literature evidence? See above, 1 and 10.

13) Even if one or more literature references are available for a species Trait – how can we be sure that its correct? –the wording in line 13 "- obligate traceability of information (every entry is linked to at least one source)" seems overemphasized (see also line 157-158). We emphasize the traceability of information because it is a quality criterion of this database. It is not possible for us to prove that the source is correct. The final choice to use this information or not lies with the user.

14) When a literature based trait is not possible to find for a species, and when a specific trait (for example "size") can vary from one geographic area to another, a "obligate literature traceability" can be misleading information. See above, 4-6.

15) Line 86-88: traits used in previous studies and databases should be favored to enable comparisons across studies (Degen et al. 2018), and 3) the traits should be usable across a wide geographical area (Bremner et al. 2006). Characteristics such as "body weight", "size", "morphology", "temperature preference", and "depth range", are area depended and subjected to adaptivity/plasticity for existing environment. Do to the "line 13 - obligate traceability of information (every entry is linked to at least one source)" meaning that your species need to be pre-defined with a trait, please explain why changes in "field-based traits" such as "size" cannot be used. See above, 4-6.

16) Field based characteristics are very important "traits" in monitoring to detect "early warning signals" (see also line 70). These type of field based traits cannot be useful if lumped into large modalities as for example S1-S5 and W1-W5 because a "catastrophe" need to happen before a species will fall from one modality to another. And if literature based evidence shall be given for a trait, such as "size" – how can we detect a change in an area? Please explain why "field-based traits" are not coming into consideration in your choice of traits (line 85-88)?

See above, 4-6.

17) Line 85-88 – says that chosen traits shall be available and applicable to all benthic taxa . .. but still . . . as also stated in the manuscript – not all species (particularly not the Arctic species) can be traitet due to the lack of literature based evidence – so what do you do when you will like to apply traits to a dataset? Does this means that the trait database cannot be used because of this lack of literature based trait evidence? Please explain how to cope with this.

See above, 1 and 10.

18) Traits must be selected in accordance to the scientific question. If a comparison to other areas are important not only same trait and modality has to be used, but also the same type of analyses. Please describe the steps from the fuzzy coding, the analyses and the mapping that can be applied to "all" in order to compare across regions.

It is not the aim of this paper to explain trait-based approaches, and it is also not according to the guidelines of ESSD, which is a data journal. In addition, there is no one single analyses. There are at least as many types of analyses that can be done with traits as can be done with species occurrence data. Every statistical analysis depends on the scientific question in mind, the only thing that changes here is the type of input data (i.e. traits instead of occurrences or biomass or abundances). Nevertheless, in order to guide interested (but yet unaware) readers towards appropriate literature we added even more references now in the introduction, line 29f and 40.

19) Line 115-148: You divide your traits into "indicators of ecosystem functions (effect traits)" and "changes in the environment (response traits)". But it is unclear how you use these two categories in table 3. Can you please mention them specifically here?

This is a common approach, not invented by us. To make this clear we refer now to an older review, Hooper et al. (2005). We decided not to use these categories in Table 3, as traits can be both, effect and response trait, depending on the situation, or neither, which might be more confusing than actually add additional value to this table. Instead we give specific examples, e.g. "Size has a direct effect on productivity ...", or "Long loved animals are more susceptible to disturbance...". To clarify this we changed the text of the table capture now to "The relation of the respective trait to benthic ecosystem functions or responses (i.e. its role as effect or response trait) are given via specific examples and underlying literature sources are displayed."

20) Please make it clear in Table 3 if Body Size, BodyWeight, Zoogeography, Depth Range are "Field-based Traits" or if they have to be "obligate traceability of information (every entry is linked to at least one source)".

See above, 4.

21) Table 3: Please define the "Zoogeography" Trait. What is an Arctic, a Arctic-boreal, and a boreal species defined as. . . what temperature ranges?

This trait relates to the biogeographic distribution (unfortunately we don't know the temperature preferences of many arctic taxa). Added the following definition into table 3 (and online):

"Definition: Spatial distribution of a species in relation to commonly used zoogeographic regions.

Arctic: Confined to Arctic regions.

Arctic-boreal: Arctic, sub-Arctic and North Atlantic/North Pacific distribution.

Boreal: North Atlantic and/or North Pacific distribution, sub-Arctic regions such as Southern Barents Sea or Bering Sea.

Cosmopolite: Cosmopolite distribution"

22) If lines 149-154 is true, please add them up front in the paper so it is clear that a "trait" have to be both "field-based" and "literature-based" in order to be able to apply a traitbased analyses on all, and not only on a subset of species.

We added this information now also in the introduction in line 41: "... labor intensive survey of literature, databases, field data, and expert knowledge."

23) Chap 4.2 – it is also obvious that the trait database most efficiently covers macrofaunal species as Annelida, Arthropoda, Mollusca .. It might be written up front in the manuscript that mega-fauna, such as Echinodermata, Sponges are "under development" and therefore not fully operative due to many missing literature based-traits.

Added "At present stage mainly species in the macrofauna size class have been uploaded" in part 2.1, line 81.

24) Line 188: I agree that "Body weight" can be removed from the database because this trait is plastic and variable within and between areas, i.e. not a global constant, and will not be covered efficiently by 1 or 2 literature based references.

We decided to remove this trait because information on max. body weight was rarely found, not because its plasticity (see comments on this topic before, 4-6).

[revised manuscript text omitted]

One common approach to use traits is as indicators of ecosystem functions (effect traits) or of changes in the environment (response traits) (Hooper et al., 2005). An overview of how each of the 19 traits that are currently included in the database may relate to ecosystem functions or respond to environmental changes or pressures is

given in Table 3.

130

**Table 3.** Detailed information on the 19 biological traits currently included in the Arctic Traits Database, clustered into morphology traits (5), life history traits (3), and behavioral traits (11). For every trait and its categories, the definition as used in the Arctic Traits Database is given. Abbreviations of each category are given (e.g. S1, S2) as these are used in files downloaded from the website. The relation of the respective trait to benthic ecosystem functions or responses (i.e. its role as effect or response trait) are given via specific examples and underlying literature sources are displayed.

**MORPHOLOGY**

| Body size  |         |                                                                                                |                      |  |  |
|------------|---------|------------------------------------------------------------------------------------------------|----------------------|--|--|
| Definition | Maxii   | Maximum body size as adult given in mm, as individual or colony and excluding appendages. Can  |                      |  |  |
|            | be hei  | leight in rather upright animals (e.g. corals), body width or diameter in rather round animals |                      |  |  |
|            | (e.g. c | crabs), or body length in elongated a                                                          | nimals (e.g. worms). |  |  |
| Categories | S1      | small                                                                                          | < 10 mm              |  |  |
|            | S2      | small-medium                                                                                   | 10-50 mm             |  |  |
|            | S3      | medium                                                                                         | 50-100 mm            |  |  |
|            | S4      | medium-large                                                                                   | 100-300 mm           |  |  |
|            | S5      | large                                                                                          | > 300 mm             |  |  |

| Body form         Definition         The external characteristic of an organism.           Categories         BF1         globulose         Round or oval (e.g. sea urchin, sponge, som           BF2         vermiform         Wormlike         Species that are flat, or encrusting (e.g. star sponge)           BF4         laterally compressed         Thin (e.g. isopods, amphipods, some bivalv BF5         upright           Function         The body form can be indicative for the ecological role of species in an ecosystem (e.g. habitat-forming), and for its vulnerability to mechanical disturbances (e.g. bottom traw Species with an upright body form will be more affected than vermiform or flat ones. S restrictions to habitat use and migration capability. Vermiform taxa can be a proxy for quality/decomposition.           Remark         Often simply a proxy of taxonomy (e.g. vermiform > polychaetes, laterally compressed amphipods).           References         Beauchard et al., 2017; Bolam and Eggleton, 2014; Costello et al., 2015; Törnroos and 2012; Wiedmann et al., 2014 Fragility Definition         The degree to which an organism can withstand physical impact.           F1         fragile         Likely to crush, break, or crack as a result of physical im brittle star, soft worms, smaller crustaceans, mollusks wit shells)           F2         intermediate         Liable to suffer minor damage, chips or racks as result or impacts or robust           F1         fragile         Likely to be damaged as a result of physical impact,                                                                                                                                                                                                                                                                                                                                                                                | Function
Detail
References | Size has a direct effect on productivity, the amount of habitat structuring and facilitation, and is important for the amount of oxygen and nutrient flux across the sediment-water interface. It correlates with food web structure, trophic levels, and energy flow in ecosystems. Smaller animals are faster growing, usually show a higher productivity and are less affected by trawling as they are more likely to fit through the net of trawling gear, thus often replacing larger slow-growing fauna in trawl-impacted areas. A clear majority of small-bodied species may be indicative for environments with high instability or be the result of environmental or anthropogenic disturbances. Larger taxa usually show a lower productivity but higher carbon fixation and have a higher effect on fluxes of nutrients, energy and matter. They usually grow slower, reproduce later, and are more affected by trawling and other disturbances. Bolam and Eggleton, 2014; Bremner, 2008; Costello et al., 2015; Emmerson, 2012; Micheli and Halpern, 2005; Norkko et al., 2013; van der Linden et al., 2016 |
|------------------------------------------------------------------------------------------------------------------------------------------------------------------------------------------------------------------------------------------------------------------------------------------------------------------------------------------------------------------------------------------------------------------------------------------------------------------------------------------------------------------------------------------------------------------------------------------------------------------------------------------------------------------------------------------------------------------------------------------------------------------------------------------------------------------------------------------------------------------------------------------------------------------------------------------------------------------------------------------------------------------------------------------------------------------------------------------------------------------------------------------------------------------------------------------------------------------------------------------------------------------------------------------------------------------------------------------------------------------------------------------------------------------------------------------------------------------------------------------------------------------------------------------------------------------------------------------------------------------------------------------------------------------------------------------------------------------------------------------------------------------------------------------------------------------------------------------------------------------------------------------------------------------------------------------------------------------------------------------------|----------------------------------|-------------------------------------------------------------------------------------------------------------------------------------------------------------------------------------------------------------------------------------------------------------------------------------------------------------------------------------------------------------------------------------------------------------------------------------------------------------------------------------------------------------------------------------------------------------------------------------------------------------------------------------------------------------------------------------------------------------------------------------------------------------------------------------------------------------------------------------------------------------------------------------------------------------------------------------------------------------------------------------------------------------------------------------------------------------------------------------------------------------------------|
| Definition       The external characteristic of an organism.         Categories       BF1       globulose       Round or oval (e.g. sea urchin, sponge, som
BF2         Operation       Wormlike       BF3       dorso-ventral compressed       Species that are flat, or encrusting (e.g. star
sponge)         BF4       laterally compressed       Thin (e.g. isopods, amphipods, some bivaly
BF5         Function       The body form can be indicative for the ecological role of species in an ecosystem (e.g.
habitat-forming), and for its vulnerability to mechanical disturbances (e.g. bottom traw
Species with an upright body form will be more affected than vermiform of flat ones. 2         Remark       Often simply a proxy of taxonomy (e.g. vermiform > polychaetes, laterally compressed
amphipods).         References       Beauchard et al., 2017; Bolam and Eggleton, 2014; Costello et al., 2015; Törnroos and
2012; Wiedmann et al., 2014         Fragility       Definition       The degree to which an organism can withstand physical impact.
F1 fragile       F1 fragile
Likely to crush, break, or crack as a result of physical im
brittle star, soft worms, smaller crustaceans, molluks wi
shells)         F2       intermediate       Likely to prush, break, or crack as a result of physical impacts, e
mough to withstand impact, or leathery or wiry en
resist impact (e.g. molluks with thicker shells, animals with f
cuicle like some echinoderms)         F3       robust       Unlikely to be damaged as a result of physical impacts, e
tough enough to withstand impact, or leathery or wiry en
resist impact (e                                                                                                                                                                                                                                                                               | Body form                        |                                                                                                                                                                                                                                                                                                                                                                                                                                                                                                                                                                                                                                                                                                                                                                                                                                                                                                                                                                                                                                                                                                                         |
| Categories       BF1       globulose       Round or oval (e.g. sea urchin, sponge, som
Wormlike         BF2       vermiform       Wormlike         BF3       dorso-ventral compressed       Species that are flat, or encrusting (e.g. star
sponge)         BF4       laterally compressed       Thin (e.g. isopods, amphipods, some bivaly
BF5         Punction       The body form can be indicative for the ecological role of species in an ecosystem (e.g.
habitat-forming), and for its vulnerability to mechanical disturbances (e.g. bottom traw
Species with an upright body form will be more affected than vermiform or flat ones. S
restrictions to habitat use and migration capability. Vermiform taxa can be a proxy for
quality/decomposition.         Remark       Often simply a proxy of taxonomy (e.g. vermiform > polychaetes, laterally compressed
amphipods).         References       Beauchard et al., 2017; Bolam and Eggleton, 2014; Costello et al., 2015; Tornroos and
2012; Wiedmann et al., 2014 Fragility Definition       The degree to which an organism can withstand physical impact.
F1         F2       intermediate       Likely to crush, break, or crack as a result of physical im
brittle star, soft worms, smaller crustaceans, mollusks wi
shells)         F2       intermediate       Likely to admaged as a result of physical impacts, e
tough enough to withstand impact, or pack as result of
miscas are stronger affected by trawling. Indicative of prey accessibility
ingestion.         F4       netermetes as stronger affected by trawling. Indicative of prey accessibi                                                                                                                                                                                                                                                                                                           | Definition                       | The external characteristic of an organism.                                                                                                                                                                                                                                                                                                                                                                                                                                                                                                                                                                                                                                                                                                                                                                                                                                                                                                                                                                                                                                                                             |
| BF4       laterally compressed       Thin (e.g., isopods, amphipods, some bivaly
E.g. coral, basket star, sponge         Function       The body form can be indicative for the ecological role of species in an ecosystem (e.g.
habitat-forming), and for its vulnerability to mechanical disturbances (e.g. bottom traw
Species with an upright body form will be more affected than vermiform of flat ones. S         Remark       Often simply a proxy of taxonomy (e.g. vermiform > polychaetes, laterally compressed
amphipods).         References       Beauchard et al., 2017; Bolam and Eggleton, 2014; Costello et al., 2015; Törnroos and
2012; Wiedmann et al., 2014         Fragility       Definition         The degree to which an organism can withstand physical impact.         F1       fragile         Likely to crush, break, or crack as a result of physical impact.         F2       intermediate         Liable to suffer minor damage, chips or cracks as result of
minates with thicker shells, animals with for
cuticle like some echinoderms)         F3       robust       Unlikely to be damaged as a result of physical impacts, or
tough enough to withstand impact, or leathery or wiry en-
resist impact (e.g. starfish, sponges, lunicates)         Function       Determines sensitivity to physical disturbance (e.g. bottom trawing) and to predatory a
Softer/fragile bodies are stronger affected by trawling. Indicative for prey accessibility
ingestion.         References       Beauchard et al., 2017; Bolam and Eggleton, 2014; Weigel et al., 2016         Skeleton                                                                                                                                                                                                                                                                                                                                        | Categories                       | BF1
BF2globulose
vermiformRound or oval (e.g. sea urchin, sponge, some bivalves)
WormlikeBF3dorso-ventral compressedSpecies that are flat, or encrusting (e.g. starfish,
sponge)                                                                                                                                                                                                                                                                                                                                                                                                                                                                                                                                                                                                                                                                                                                                                                                                                                                                                                                            |
| Remark       Often simply a proxy of taxonomy (e.g. vermiform > polychaetes, laterally compressed amphipods).         References       Beauchard et al., 2017; Bolam and Eggleton, 2014; Costello et al., 2015; Törnroos and 2012; Wiedmann et al., 2014         Fragility       Definition       The degree to which an organism can withstand physical impact.         F1       fragile       Likely to crush, break, or crack as a result of physical im brittle star, soft worms, smaller crustaceans, mollusks wit shells)         F2       intermediate       Liable to suffer minor damage, chips or cracks as result of impacts (e.g. mollusks wit shells)         F3       robust       Unlikely to be damaged as a result of physical impact, e cough enough to withstand impact, or leathery or wiry en resist impact (e.g. bottom trawling) and to predatory a Softer/fragile bodies are stronger affected by trawling. Indicative for prey accessibility ingestion.         References       Beauchard et al., 2017; Bolam and Eggleton, 2014; Weigel et al., 2016         Skeleton       Definition       Presence and type of supporting structures in the animal body.         Categories       SK1       calcareous       Skeleton material argonite or calcite (e.g. silceous sponges)         SK3       cuticle       No skeleton material chin (e.g. anthropods)       SK4       cuticle (e.g. sea slugs)         Function       Presence and type of supporting structures in the animal body.       Sk2       silceton       Skeleton material argonite or calcite (e.g. sluc                                                                                                                                                                                                                                                                                                                                                                                            | Function                         | BF4       laterally compressed       Thin (e.g. isopods, amphipods, some bivalves)         BF5       upright       E.g. coral, basket star, sponge         The body form can be indicative for the ecological role of species in an ecosystem (e.g. if it is habitat-forming), and for its vulnerability to mechanical disturbances (e.g. bottom trawling).         Species with an upright body form will be more affected than vermiform or flat ones. Sets restrictions to habitat use and migration capability. Vermiform taxa can be a proxy for litter quality/decomposition.                                                                                                                                                                                                                                                                                                                                                                                                                                                                                                                                     |
| amphipods).       References       Beauchard et al., 2017; Bolam and Eggleton, 2014; Costello et al., 2015; Törnroos and 2012; Wiedmann et al., 2014         Fragility       Definition       The degree to which an organism can withstand physical impact.
F1       fragile       Likely to crush, break, or crack as a result of physical im brittle star, soft worms, smaller crustaceans, mollusks wi shells)         F2       intermediate       Liable to suffer minor damage, chips or cracks as result of impacts (e.g. mollusks with thicker shells, animals with f cruticle like some echinoderms)         F3       robust       Unlikely to be damaged as a result of physical impacts, e tough enough to withstand impact, or leathery or wiry en resist impact (e.g. starfish, sponges, tunicates)         Function       Determines sensitivity to physical disturbance (e.g. bottom trawling) and to predatory a Softer/fragile bodies are stronger affected by trawling. Indicative for prey accessibility ingestion.         References       Beauchard et al., 2017; Bolam and Eggleton, 2014; Weigel et al., 2016         Skeleton       Definition         Presence and type of supporting structures in the animal body.         Categories       SK1         SK3       chitnous         SK4       cuticle         No form of protective structure (e.g. sea slugs)         Function       Presence and type of supporting value and init (e.g. attripopods)         SK4       cuticle       No skeleton material argonite or calcite (e.g.                                                                                                                                                                                                                                                                                                                                                                                                                           | Remark                           | Often simply a proxy of taxonomy (e.g. vermiform > polychaetes, laterally compressed >                                                                                                                                                                                                                                                                                                                                                                                                                                                                                                                                                                                                                                                                                                                                                                                                                                                                                                                                                                                                                                  |
| Fragility         Definition       The degree to which an organism can withstand physical impact.
F1       fragile       Likely to crush, break, or crack as a result of physical im
brittle star, soft worms, smaller crustaceans, mollusks wi
shells)         F2       intermediate       Liable to suffer minor damage, chips or cracks as result of
impacts (e.g. mollusks with thicker shells, animals with th
cuticle like some echinoderms)         F3       robust       Unlikely to be damaged as a result of physical impact, e
tough enough to withstand impact, or leathery or wiry en
resist impact (e.g. starfish, sponges, tunicates)         Function       Determines sensitivity to physical disturbance (e.g. bottom trawling) and to predatory a
Softer/fragile bodies are stronger affected by trawling. Indicative for prey accessibility
ingestion.         References       Beauchard et al., 2017; Bolam and Eggleton, 2014; Weigel et al., 2016         Skeleton       SK1         Definition       Presence and type of supporting structures in the animal body.         Categories       SK1         SK4       cuticle         No skeleton material arigonite or calcite (e.g. bivalves)
SK2         SK4       cuticle         No form of protective structure like a cuticle (e.g. soft
or palatability), and cosystem engineering (provision of habitat, increased heterogeneity)
calcifying taxa contribute most to inorganic carbon sequestration.         Costability       Definition       The degree to which species aggregate.                                                                                                                                                                                                                                                                                                                                                                                      | References                       | amphipods).
Beauchard et al., 2017; Bolam and Eggleton, 2014; Costello et al., 2015; Törnroos and Bonsdorff, 2012; Wiedmann et al., 2014                                                                                                                                                                                                                                                                                                                                                                                                                                                                                                                                                                                                                                                                                                                                                                                                                                                                                                                                                                             |
| Definition       The degree to which an organism can withstand physical impact.         F1       fragile       Likely to crush, break, or crack as a result of physical impact.         F2       intermediate       Liable to suffer minor damage, chips or cracks as result of impacts (e.g. mollusks with thicker shells, animals with fouriele like some echinoderms)         F3       robust       Unlikely to be damaged as a result of physical impacts, et ough enough to withstand impact, or leathery or wiry en resist impact (e.g. starfish, sponges, tunicates)         Function       Determines sensitivity to physical disturbance (e.g. bottom trawling) and to predatory a Softer/fragile bodies are stronger affected by trawling. Indicative for prey accessibility ingestion.         References       Beauchard et al., 2017; Bolam and Eggleton, 2014; Weigel et al., 2016         Skeleton       Definition         Presence and type of supporting structures in the animal body.         Categories       SK1         SK2       siliceous         Skeleton material ariagonite or calcite (e.g. bivalves)         SK2       siliceous         SK4       cuticle         No form of protective structure like a cuticle (e.g. sa slugs)         Function       Indicates vulnerability (trawling, ocean acidification), resistance to predation (proxy of palatability), and ecosystem engineering (provision of habitat, increased heterogeneity) calcifying taxa contribute most to inorganic carbon sequestration.                                                                                                                                                                                                                                                                                                                                                                                                                                                  | Fragility                        |                                                                                                                                                                                                                                                                                                                                                                                                                                                                                                                                                                                                                                                                                                                                                                                                                                                                                                                                                                                                                                                                                                                         |
| Function       Determines sensitivity to physical disturbance (e.g. bottom trawling) and to predatory a Softer/fragile bodies are stronger affected by trawling. Indicative for prey accessibility ingestion.         References       Beauchard et al., 2017; Bolam and Eggleton, 2014; Weigel et al., 2016         Skeleton       Definition         Definition       Presence and type of supporting structures in the animal body.         Categories       SK1         calcareous       Skeleton material aragonite or calcite (e.g. bivalves)         SK2       siliceous         Skeleton material silicate (e.g. siliceous sponges)         SK3       chitinous         Skeleton material chitin (e.g. arthropods)         SK4       cuticle         No skeleton but a protective structure (e.g. sea slugs)         Function       Indicates vulnerability (trawling, ocean acidification), resistance to predation (proxy of palatability), and ecosystem engineering (provision of habitat, increased heterogeneity) calcifying taxa contribute most to inorganic carbon sequestration.         References       Costello et al., 2015; Frid and Caswell, 2016, 2015; Spitz et al., 2014         Sociability       Definition         The degree to which species aggregate.       Single individual         Categories       SO1       solitary         Single individuals forming groups; growing in clusters (e biorder)                                                                                                                                                                                                                                                                                                                                                                                                                                                                                                                                                         | Definition                       | F1fragileLikely to crush, break, or crack as a result of physical impact (e.g.
brittle star, soft worms, smaller crustaceans, mollusks with thin
shells)F2intermediateLiable to suffer minor damage, chips or cracks as result of physical
impacts (e.g. mollusks with thicker shells, animals with harder
cuticle like some echinoderms)F3robustUnlikely to be damaged as a result of physical impacts, e.g. hard or
tough enough to withstand impact, or leathery or wiry enough to
resist impact (e.g. starfish sponges tunicates)                                                                                                                                                                                                                                                                                                                                                                                                                                                                                                                                                                 |
| References       Beauchard et al., 2017; Bolam and Eggleton, 2014; Weigel et al., 2016         Skeleton       Definition       Presence and type of supporting structures in the animal body.         Categories       SK1       calcareous       Skeleton material aragonite or calcite (e.g. bivalves)         SK2       siliceous       Skeleton material aragonite or calcite (e.g. bivalves)         SK2       siliceous       Skeleton material aragonite or calcite (e.g. bivalves)         SK3       chitinous       Skeleton material aragonite or calcite (e.g. bivalves)         SK4       cuticle       No skeleton material chitin (e.g. arthropods)         SK4       cuticle       No skeleton but a protective structure like a cuticle (e.g. sea Slugs)         Function       Indicates vulnerability (trawling, ocean acidification), resistance to predation (proxy of palatability), and ecosystem engineering (provision of habitat, increased heterogeneity) calcifying taxa contribute most to inorganic carbon sequestration.         References       Costello et al., 2015; Frid and Caswell, 2016, 2015; Spitz et al., 2014         Sociability       The degree to which species aggregate.         Categories       SO1       solitary         Single individuals       Single individuals forming groups; growing in clusters (e bivartite)                                                                                                                                                                                                                                                                                                                                                                                                                                                                                                                                                                                                                     | Function                         | Determines sensitivity to physical disturbance (e.g. bottom trawling) and to predatory aggression.
Softer/fragile bodies are stronger affected by trawling. Indicative for prey accessibility and ease of ingestion.                                                                                                                                                                                                                                                                                                                                                                                                                                                                                                                                                                                                                                                                                                                                                                                                                                                                                                 |
| Skeleton         Definition       Presence and type of supporting structures in the animal body.         Categories       SK1       calcareous       Skeleton material aragonite or calcite (e.g. bivalves)         SK2       siliceous       Skeleton material silicate (e.g. siliceous sponges)         SK3       chitinous       Skeleton material silicate (e.g. arthropods)         SK4       cuticle       No skeleton but a protective structure like a cuticle (e.g. s         SK5       none       No form of protective structure (e.g. sea slugs)         Function       Indicates vulnerability (trawling, ocean acidification), resistance to predation (proxy of palatability), and ecosystem engineering (provision of habitat, increased heterogeneity) calcifying taxa contribute most to inorganic carbon sequestration.         References       Costello et al., 2015; Frid and Caswell, 2016, 2015; Spitz et al., 2014         Sociability       The degree to which species aggregate.         Categories       SO1       solitary         Single individual       Sorea proving in clusters (et al., 202                                                                                                                                                                                                                                                                                                                                                                                                                                                                                                                                                                                                                                                                                                                                                                                                                                                                | References                       | Beauchard et al., 2017; Bolam and Eggleton, 2014; Weigel et al., 2016                                                                                                                                                                                                                                                                                                                                                                                                                                                                                                                                                                                                                                                                                                                                                                                                                                                                                                                                                                                                                                                   |
| Definition
CategoriesPresence and type of supporting structures in the animal body.CategoriesSK1calcareousSkeleton material aragonite or calcite (e.g. bivalves)SK2siliceousSkeleton material silicate (e.g. siliceous sponges)SK3chitinousSkeleton material chitin (e.g. arthropods)SK4cuticleNo skeleton but a protective structure like a cuticle (e.g. sFunctionIndicates vulnerability (trawling, ocean acidification), resistance to predation (proxy of palatability), and ecosystem engineering (provision of habitat, increased heterogeneity)
calcifying taxa contribute most to inorganic carbon sequestration.ReferencesCostello et al., 2015; Frid and Caswell, 2016, 2015; Spitz et al., 2014Sociability
SO2Single individual
SO2SO2gregariousSingle individuals forming groups; growing in clusters (e                                                                                                                                                                                                                                                                                                                                                                                                                                                                                                                                                                                                                                                                                                                                                                                                                                                                                                                                                                                                                                                                                                                                                              | Skeleton                         |                                                                                                                                                                                                                                                                                                                                                                                                                                                                                                                                                                                                                                                                                                                                                                                                                                                                                                                                                                                                                                                                                                                         |
| Function       Indicates vulnerability (trawling, ocean acidification), resistance to predation (proxy of palatability), and ecosystem engineering (provision of habitat, increased heterogeneity) calcifying taxa contribute most to inorganic carbon sequestration.         References       Costello et al., 2015; Frid and Caswell, 2016, 2015; Spitz et al., 2014         Sociability       Definition         Categories       SO1 solitary         Single individual       Single individuals forming groups; growing in clusters (et al., 2015)                                                                                                                                                                                                                                                                                                                                                                                                                                                                                                                                                                                                                                                                                                                                                                                                                                                                                                                                                                                                                                                                                                                                                                                                                                                                                                                                                                                                                                        | Definition
Categories         | Presence and type of supporting structures in the animal body.SK1calcareousSkeleton material aragonite or calcite (e.g. bivalves)SK2siliceousSkeleton material silicate (e.g. siliceous sponges)SK3chitinousSkeleton material chitin (e.g. arthropods)SK4cuticleNo skeleton but a protective structure like a cuticle (e.g. sea-squirts)SK5noneNo form of protective structure (e.g. sea slugs)                                                                                                                                                                                                                                                                                                                                                                                                                                                                                                                                                                                                                                                                                                                         |
| References       Costello et al., 2015; Frid and Caswell, 2016, 2015; Spitz et al., 2014         Sociability       Definition       The degree to which species aggregate.         Categories       SO1 solitary       Single individual solitary single individual solitary solitary single individual solitary so | Function                         | Indicates vulnerability (trawling, ocean acidification), resistance to predation (proxy of palatability), and ecosystem engineering (provision of habitat, increased heterogeneity). Large calcifying taxa contribute most to inorganic carbon sequestration.                                                                                                                                                                                                                                                                                                                                                                                                                                                                                                                                                                                                                                                                                                                                                                                                                                                           |
| Sociability           Definition         The degree to which species aggregate.           Categories         SO1 solitary         Single individual           SO2         gregarious         Single individuals forming groups; growing in clusters (et al. and the species)                                                                                                                                                                                                                                                                                                                                                                                                                                                                                                                                                                                                                                                                                                                                                                                                                                                                                                                                                                                                                                                                                                                                                                                                                                                                                                                                                                                                                                                                                                                                                                                                                                                                                                                   | References                       | Costello et al., 2015; Frid and Caswell, 2016, 2015; Spitz et al., 2014                                                                                                                                                                                                                                                                                                                                                                                                                                                                                                                                                                                                                                                                                                                                                                                                                                                                                                                                                                                                                                                 |
| Definition
CategoriesThe degree to which species aggregate.SO1solitarySingle individualSO2gregariousSingle individuals forming groups; growing in clusters (e                                                                                                                                                                                                                                                                                                                                                                                                                                                                                                                                                                                                                                                                                                                                                                                                                                                                                                                                                                                                                                                                                                                                                                                                                                                                                                                                                                                                                                                                                                                                                                                                                                                                                                                                                                                                                               | Sociability                      |                                                                                                                                                                                                                                                                                                                                                                                                                                                                                                                                                                                                                                                                                                                                                                                                                                                                                                                                                                                                                                                                                                                         |
| SO3 colonial barnacles)
Living in permanent colonies (e.g. stony corals, Bryozoa,
Synascidia)                                                                                                                                                                                                                                                                                                                                                                                                                                                                                                                                                                                                                                                                                                                                                                                                                                                                                                                                                                                                                                                                                                                                                                                                                                                                                                                                                                                                                                                                                                                                                                                                                                                                                                                                                                                                                                                                                            | Definition
Categories         | The degree to which species aggregate.SO1solitarySingle individualSO2gregariousSingle individuals forming groups; growing in clusters (e.g.
barnacles)SO3colonialLiving in permanent colonies (e.g. stony corals, Bryozoa,
Synascidia)                                                                                                                                                                                                                                                                                                                                                                                                                                                                                                                                                                                                                                                                                                                                                                                                                                                                            |

[revised manuscript text omitted]

---

## Referee Report (RR1)

Authors have made several changes and improvements, thanks. For most of my comments on the original manuscript, however, they responded in perfunctory or defensive manner. Fair enough, they represent current expert practitioners while this reviewer has not sorted a benthic sample in more than three decades. I evaluate the revised manuscript according to helpful guidelines posted by ESSD, e.g. at https://www.earth-syst-sci-data.net/10/2275/2018/.

I like this statement from those guidelines "effective evaluation should encourage authors to modify their initial submission in directions and with amendments that allow the data product as eventually published to more closely meet the full range of recommendations". Have these authors made sufficient modifications in order to better meet those guidelines?

1. **Open access** - remains deficient, fails to meet this standard: "Users should not encounter registration steps, password requests, access agreements, or other log-in barriers or tracking mechanisms." Not fatal but not exactly in compliance. Other repositories, e.g. CEH where terrestrial landscape data from Woods et al. reside, impose a registration step. Why do ecologists seem to hide behind these registration barriers? For a fully free and open example, look at ESSD-2019-17. This reviewer not impressed nor reassured by the justification given and the supposed confidentiality protections. Follow other good ESSD examples and just put the information in full open un-restricted access - no good reason why not! Not a reason for disqualification but something that the authors could improve.
2. **Permanent identifiers** - okay, no issues.
3. **Useful data descriptions including source attribution** - good.
4. **Codes and tools** - now available at the figshare link.
5. **Uncertainty analysis** - missing, but perhaps not essential or even appropriate for a database at the start-up stage? Again look at ESSD-2019-17 for an example of a more developed (15 years!) community database which has reached the stage where it can expose valid community-recognized uncertainties. These authors provide some uncertainty hints in the Discussion section where they list under-representation (a serious weakness?), divergence in trait descriptions (what they call 'conflicts'), and sampling biases. But nowhere do we get a sense of overall uncertainty, today or as hoped for in the future?
6. **Data availability** - adequate, happy to see the university take a prominent role in data archiving but not quite to standard for open access reasons already mentioned.
7. **Interest and utility**, with this goal "*ensure that ESSD products enable substantial advances in future research.*" These authors list some 'substantial advance' goals: "increase our ecological understanding of this rapidly changing [Arctic] system" and "a cutting-edge tool for (not only) the marine realm and a role-model for prospective databases". Unfortunately, after details of data base construction and content, they seem to have relaxed to the (more realistic?) goal of "tool for collecting and providing information". They raised our hopes for a useful ecological analysis breakthrough but have so far gotten only as far as a (seriously incomplete?) trait accounting system? They need to lower expectations for the short term but clarify hopes for the longer term? I wonder what this product would look like in 5 years? Will it have stimulated a more systematic approach to Arctic benthic research? Will it then serve a larger community of researchers? Will it enable research on carbon or nutrient cycling, ecosystem function, ecosystem change, etc., issues that will prove crucial for the future Arctic? I take the point that publication in ESSD could stimulate interest and use, but the product as described seems highly preliminary and tentative, without - as yet - clear demonstration of community buy-in or broad research applicability. Again, the international gravity consortium database example (ESSD-2019-17) - perhaps not a fair comparison! - seems at least to point in some directions that this product might like to go? I miss a realistic calibrated message from the authors that distinguishes what they have (or have yet to) achieve versus what they hope as the eventual impact of this product.